# Public-data Assisted Private Stochastic Optimization: Power and Limitations

**Enayat Ullah**
Meta *
enayat@meta.com

**Michael Menart**
Department of Computer Science & Engineering
The Ohio State University [†]
Department of Computer Science, University of Toronto
Vector Institute
menart.2@osu.edu

**Raef Bassily**
Department of Computer Science & Engineering
Translational Data Analytics Institute (TDAI)
The Ohio State University
bassily.1@osu.edu

**Cristóbal Guzmán**
Inst. for Mathematical and Comput. Eng.
Fac. de Matemáticas and Esc. de Ingeniería
Pontificia Universidad Católica de Chile
crguzmanp@uc.cl

**Raman Arora**
Department of Computer Science
Johns Hopkins University
arora@cs.jhu.edu

## Abstract

We study the limits and capability of public-data assisted differentially private (PA-DP) algorithms. Specifically, we focus on the problem of stochastic convex optimization (SCO) with either labeled or unlabeled public data. For complete/labeled public data, we show that any $(\epsilon, \delta)$-PA-DP has excess risk $\tilde{\Omega}\big(\min\big\{\frac{1}{\sqrt{n_{\text{pub}}}}, \frac{1}{\sqrt{n}} + \frac{\sqrt{d}}{n\epsilon}\big\}\big)$, where $d$ is the dimension, $n_{\text{pub}}$ is the number of public samples, $n_{\text{priv}}$ is the number of private samples, and $n = n_{\text{pub}} + n_{\text{priv}}$. These lower bounds are established via our new lower bounds for PA-DP mean estimation, which are of a similar form. Up to constant factors, these lower bounds show that the simple strategy of either treating all data as private or discarding the private data, is optimal. We also study PA-DP supervised learning with *unlabeled* public samples. In contrast to our previous result, we here show novel methods for leveraging public data in private supervised learning. For generalized linear models (GLM) with unlabeled public data, we show an efficient algorithm which, given $\tilde{O}(n_{\text{priv}}\epsilon)$ unlabeled public samples, achieves the dimension independent rate $\tilde{O}\big(\frac{1}{\sqrt{n_{\text{priv}}}} + \frac{1}{\sqrt{n_{\text{priv}}\epsilon}}\big)$. We develop new lower bounds for this setting which shows that this rate cannot be improved with more public samples, and any fewer public samples leads to a worse rate. Finally, we provide extensions of this result to general hypothesis classes with finite *fat-shattering dimension* with applications to neural networks and non-Euclidean geometries.

---

[*]Work done while the author was at the Johns Hopkins University.

[†]This work was done while M. Menart was at The Ohio State University.

38th Conference on Neural Information Processing Systems (NeurIPS 2024).

# 1  Introduction

The framework of differential privacy has become the primary standard for protecting individual privacy in data analysis and machine learning. Unfortunately, this rigorous framework has also been shown to lead to worse performance on such tasks both empirically and in theory [BST14, PVX⁺23]. However, it is often the case that, in addition to a collection of privacy-sensitive data points, analysts have access to a pool of public data, for which guaranteeing privacy protections is not required. This can happen, for example, when consumers deem their own data non-sensitive and opt-in to sell this data to a company. This has motivated a long line of work analyzing how public data can be leveraged in tandem with private data to provide better utility [BNS13, ABM19, BCM⁺20, ZWB21, BKS22, AGM⁺22, NMT⁺23]. In machine learning. for example, two commonly proposed strategies are public pretraining and using public data to identify gradient subspaces [ZWB21, KDRT21]. Public pretraining, in particular, has proven effective in practice [YNB⁺22a, BWZK22], and prior work has even identified a *specific* problem instance where public and private data used in tandem leads to better rates than is possible using only the public or private datasets in isolation [GHN⁺23]. Despite this surge of work, theory has struggled to show that public data leads to fundamental rate improvements more generally. Recent work has even shown that, for the problem of pure PA-DP stochastic convex optimization, a small amount of public data, $n_{\text{pub}} \leq n\epsilon/d$, leads to no rate improvement, where $n = n_{\text{pub}} + n_{\text{priv}}$ and $n_{\text{priv}}$ is the number of private samples [LLHR23].

One particularly important version of this problem is in supervised learning when the public data is unlabeled. This setting has found importance in medical domains and deep learning more generally [LW19, SCZ⁺20, PAE⁺17]. Notably, unlabeled data is much less time intensive to collect than labeled data. Due to this fact, and the fact that the unlabeled public data does not contain the same kind of information contained in the private data, the regime $n_{\text{pub}} = \Omega(n_{\text{priv}})$ is meaningful both in theory and in practice. We also note this setting is a stronger (in terms of privacy) version of the label-private setting, where only the labels of the dataset are considered private [CH11, BNS13].

Motivated by the importance of these settings and the lack of existing theory for them in stochastic optimization, we study fundamental limitations and applications of public data in $(\epsilon, \delta)$-PA-DP stochastic optimization. In the case where the public data is complete/labeled, we show that the application of public data is fundamentally limited. We then contrast this result with new results in the unlabeled public data setting. In this setting, we provide new results for GLMs, and extend these results to more general hypothesis classes, with finite fat-shattering dimension, and non-Euclidean geometries.

## 1.1  Our Contributions
We outline our primary contributions in the following.

**Limits of Private Stochastic Convex Optimization with Public Data.** First, we show a tight lower bound for the problem of differentially-private stochastic convex optimization (DP-SCO) assisted with complete public data, that is, the public data and private data have the same number of features (and labels when applicable). Specifically, we show a lower bound of $\Omega\Big( \min\big\{ \frac{1}{\sqrt{n_{\text{pub}}}}, \frac{1}{\sqrt{n}} + \frac{\sqrt{d}}{n\epsilon} \big\} \Big)$ on the excess population risk for this problem. When $d \geq n\epsilon$ and $n_{\text{pub}} \leq \frac{n}{\log(1/\delta)}$, we further improve this lower bound to $\Omega\Big( \min\big\{ \frac{1}{\sqrt{n_{\text{pub}}}}, \frac{1}{\sqrt{n}} + \frac{\sqrt{d\log(1/\delta)}}{n\epsilon} \big\} \Big)$. This lower bound is matched by the simple upper bound strategy which either discards the private data entirely and outputs the public mean or simply treats all data-points as private. Barring constant factors, this shows more sophisticated attempts at leveraging public data will yield no benefit. These results also hold even for generalized linear models. Our results are based on new results we establish for DP mean estimation with public data, and a reduction of mean estimation to SCO. We note that previous work [LLHR23], on this problem either focused on the pure PA-DP case when $n_{\text{pub}} \leq n\epsilon/d$, or, in the approximate PA-DP case, did not obtain the dimension dependence. Our mean estimation lower bound uses a novel analysis of fingerprinting codes [BUV14], and our SCO reduction further builds on ideas from [BST14, CWZ21]. We also show that, when $d \geq n\epsilon$, our lower bounds for approximate PA-DP SCO directly imply a tight lower bound for *pure* PA-DP.

**Private Supervised Learning with Unlabeled Public Data.** While the previously discussed results show there is no hope for leveraging public data in "interesting" ways, even for GLMs, they do not preclude settings where the public data is less informative. In particular, in the setting where the

public data is *unlabeled*, it makes sense to even consider $n_{\text{pub}} \geq n_{\text{priv}}$. In this setting, we provide the following results.

- For (Euclidean) GLMs we develop an efficient algorithm which, given $\tilde{O}(n_{\text{priv}}\epsilon)$ unlabeled public data points, achieves the dimension independent rate $\tilde{O}\big(\frac{1}{\sqrt{n_{\text{priv}}}} + \frac{1}{\sqrt{n_{\text{priv}}\epsilon}}\big)$. We obtain this result via a dimensionality reduction procedure of the private feature vectors using the public data, and then running an efficient private algorithm in the lower dimensional space. The key idea is that public data can be used to identify a low dimensional subspace, which under the appropriate metric acts as a cover for the higher dimensional space. We elucidate the tightness of our upper bound by proving two new lower bounds which show that access to a greater number of unlabeled public samples cannot improve this rate, and that any fewer public samples lead to a worse rate. While dimension independent rates for the GLMs have previously been developed in the *unconstrained* setting [SSTT21, ABG⁺22], in the constrained setting which we study, dependence on dimension is known to be unavoidable even for GLMs if no public data is available [BST14]. Our result thus allows us to bypass these limitations.

- By observing that the key requirement in our GLM result is the construction of an appropriate cover, we extend this result to general hypothesis classes with bounded *fat-shattering dimension*. In the non-private setting, it is known that finiteness of fat-shattering dimension characterizes learnability of real-valued predictors with *scale-sensitive* losses [BLW94, ABDCBH97]. In the private setting, such a result is not known, and is in fact impossible in the *proper learning* setting. This follows from the fact that norm bounded linear predictors, regardless of their (ambient) dimension $d$, have the same fat-shattering dimension [SST10]. However, it is known that they are not learnable privately in high dimensions $d \geq (n\epsilon)^2$ [BST14]. In contrast, in the PA-DP setting, we show that it is possible to properly learn such classes with a rate of roughly $O\left(\mathfrak{R}_{n_{\text{priv}}}(\mathcal{H}) + \inf_{\alpha>0}\left(\frac{\text{fat}_\alpha(\mathcal{H})}{n_{\text{priv}}\epsilon} + \alpha\right)\right)$, where $\mathfrak{R}_{n_{\text{priv}}}(\mathcal{H})$ denotes the Rademacher complexity of $\mathcal{H}$ and $\text{fat}_\alpha(\mathcal{H})$ denotes its fat-shattering dimension at scale $\alpha$ (see Section 2 for preliminaries).

- As applications of our result for hypothesis classes with bounded fat-shattering dimension, we obtain guarantees for learning feed-forward neural networks and non-Euclidean GLMs. In particular, for depth $M$ feed-forward neural networks with weights bounded as $\|W_j\|_F \leq R_j$ and 1-Lipschitz positive homogeneous activation, we achieve an excess risk bound of essentially $\tilde{O}\left(\frac{\sqrt{M}\prod_{j=1}^{M} R_j}{\sqrt{n_{\text{priv}}}} + \left(\frac{M\left(\prod_{j=1}^{M} R_j\right)^2}{n_{\text{priv}}\epsilon}\right)^{1/3}\right)$. For non-Euclidean GLMs, our guarantees are dimension-independent which is not known to be achievable, as of yet, even in the unconstrained setting with no public data (unlike Euclidean GLMs).

## 1.2 Related Work

With regards to labeled public data, the most directly related work to ours is the recent work of [LLHR23]. This work proves a lower bound of $\Omega\left(\min\left\{\frac{1}{\sqrt{n_{\text{pub}}}}, \frac{1}{\sqrt{n}} + \frac{1}{n\epsilon}\right\}\right)$ for approximate PA-DP mean-estimation/SCO. We note that our results for approximate PA-DP crucially obtain a dependence on $d$ that is the key "price" paid for privacy in this setting. [LLHR23] also show a lower bound of $\Omega\left(\min\left\{\frac{1}{\sqrt{n_{\text{pub}}}}, \frac{1}{\sqrt{n}} + \frac{d}{n\epsilon}\right\}\right)$ on a *pure* PA-DP mean estimation/SCO, but this result only holds when $d \leq \frac{n\epsilon}{n_{\text{pub}}}$. As such, their result is orthogonal to our result in the pure PA-DP setting, which operates in the regime $d \geq n\epsilon$. In both cases, our proof technique is fundamentally different than theirs.[3] Tangentially, [BKS22] showed a small amount of public data is useful in pure-DP mean estimation when the range parameters on the data are unknown.

An important setting where public data is shown to be useful is *PAC learning*. Non-privately, it is known that the finiteness of *VC dimension* characterizes learnability [VC71, BEHW89]. However, under DP, it is impossible to PAC learn even the class of *thresholds*, which has VC dimension of one [BNS13]. The works of [BNS13, BTGT18, ABM19] showed that given access to a small unlabelled public data, it is possible to go beyond this limitation and privately learn VC classes, essentially by reducing a hypothesis class with finite VC dimension to a finite hypothesis class

---

[3] We note that concurrently and independently, version 2 of [LLHR23], [LLHR24], obtained a lower bound of $\Omega\left(\min\left\{\frac{1}{\sqrt{n_{\text{pub}}}}, \frac{1}{\sqrt{n}} + \frac{\sqrt{d}}{n\epsilon}\right\}\right)$, but their lower bound is limited to *symmetric* procedures.

A number of works have studied the impact of public data in applied settings as well. A common technique is to use public data to reduce the problem dimension in some way [ZWB21, YZCL21, PHYS24]. The work of [GHN+23] identified a specific problem instance which supports the method of public pretraining commonly used in practice.

With regards to unlabelled public data, there are several existing works. Transfer learning is a common approach in this setting. Besides the benefits in PAC learning, this setting also has applications in deep learning, where (empirically) unlabeled public data has been used to obtain performance improvements [PAE+17, PSM+18]. Unlabeled public data has also yielded impressive results used for pre-training large language models [LTLH22, YNB+22b]. We also remark that, in practice, it is reasonable to expect the private and public datasets to come from slightly different distributions. Accounting for this distribution shift has also been the study of several recent works [BKS22, BDBC+23]. However, in this work we focus on first characterizing the more fundamental problem where the public and private datasets are drawn i.i.d. from the same distribution.

## 2 Preliminaries

Here, we describe the concepts and assumptions used in the rest of this paper. In this work, $\|\cdot\|$ always denotes the $\ell_2$ norm unless stated otherwise.

**Public-Data Assisted Differential Privacy.** We first present the traditional notion of differential privacy (DP). Let $n, d \in \mathbb{N}$ and $\mathcal{X}$ be some data domain. When no public data is present, we say that an algorithm $\mathcal{A}$ satisfies $(\epsilon, \delta)$-differential privacy (DP) if for all datasets $S$ and $S'$ differing in one data point and all events $\mathcal{E}$ in the range of $\mathcal{A}$, $\mathbb{P}[\mathcal{A}(S) \in \mathcal{E}] \leq e^\epsilon \mathbb{P}[\mathcal{A}(S') \in \mathcal{E}] + \delta$ [DMNS06].

In our work, we denote the number of public samples in the dataset, $S = (S_{\text{pub}}, S_{\text{priv}}) \in \mathcal{X}^n$, as $n_{\text{pub}}$ and the number of private samples as $n_{\text{priv}}$, such that $n = n_{\text{pub}} + n_{\text{priv}}$. In keeping with previous work [BNS13, BCM+20], we define public data assisted differentially private algorithms in the following way [4].

**Definition 1** (PA-DP). *An algorithm $\mathcal{A}$ is $(\epsilon, \delta)$ public-data assisted differentially private (PA-DP) algorithm with public sample size $n_{pub}$ and private sample size $n_{priv}$ if for any public dataset $S_{pub} \in \mathcal{X}^{n_{pub}}$, and any pair of private datasets $S_{priv}, S'_{priv} \in \mathcal{X}^{n_{priv}}$ differing in at most one entry, it holds for any event $\mathcal{E}$ that $\mathbb{P}[\mathcal{A}(S_{pub}, S_{priv}) \in \mathcal{E}] \leq e^\epsilon \mathbb{P}[\mathcal{A}(S_{pub}, S'_{priv}) \in \mathcal{E}] + \delta$. When $\delta = 0$, we refer to this notion as pure PA-DP, denoted as $\epsilon$-PA-DP.*

**Stochastic Convex Optimization** Let $\mathcal{D}$ be a distribution supported on $\mathcal{X}$. Given some constraint set $\mathcal{W} \subseteq \mathbb{R}^d$ of diameter at most $D$, and a $G$ Lipschitz convex loss $\ell : \mathcal{W} \times \mathcal{X} \to \mathbb{R}$, we are interested in minimizing the population loss, $L(w; \mathcal{D}) = \mathbb{E}_{x \sim \mathcal{D}} [\ell(w; x)]$. Denote the minimizer as $w^* = \min_{w \in \mathcal{W}} \{L(w; \mathcal{D})\}$. We evaluate the quality of the approximate solution, $w$, via the excess risk, $L(w; \mathcal{D}) - L(w^*; \mathcal{D})$. Specifically, we are interested in PA-DP algorithms which minimizes this quantity when given $S_{\text{pub}}, S_{\text{priv}} \overset{i.i.d.}{\sim} \mathcal{D}$. For a datset $S$ we also define the empirical loss $\widehat{L}(w; S) = \frac{1}{|S|} \sum_{x \in S} \ell(w; x)$.

**Supervised Learning and Generalized Linear Models (GLMs)** In the supervised learning setting, in addition to the feature space $\mathcal{X}$, we define the label space $\mathcal{Y}$. We here let $\mathcal{D}$ be a joint probability distribution over $\mathcal{X} \times \mathcal{Y}$ and $\mathcal{D}_\mathcal{X}$ and $\mathcal{D}_\mathcal{Y}$ denote the respective marginal distributions. Let $\mathcal{H} \subseteq \mathbb{R}^\mathcal{X}$ be a hypothesis class of real-valued predictors, and let $\text{fat}_\alpha(\mathcal{H})$ denote its fat shattering dimension at scale $\alpha$. Consider the loss function $\ell : \mathcal{H} \times \mathcal{X} \times \mathcal{Y} \to \mathbb{R}$, such that $\ell(h; x, y) = \phi_y(h(x))$ for some function $\phi_y$. We assume that the map $\phi_y : \mathbb{R} \to \mathbb{R}$ is $G$-Lipschitz for all $y \in \mathcal{Y}$ and is $B$-bounded. Further, we assume that $\sup_{x \in \mathcal{X}} |h(x)| \leq R$ and define $\sup_{x \in \mathcal{X}} \|x\| = \|\mathcal{X}\|$.

GLMs are a special case of supervised learning setting where the hypothesis class is that of linear predictors, $\mathcal{H} = \mathcal{W} \subseteq \mathbb{R}^d$, over $\mathcal{X} \subseteq \mathbb{R}^d$, and $h(x) = w^\top x$. We refer to the public dataset of unlabeled feature vectors as $X_{\text{pub}}$.

**Covering numbers, fat-shattering and Rademacher Complexity** Given $X = (x_1, x_2, \cdots, x_m)$ the $\ell_p$ distance between two hypothesis $h_1, h_2 \in \mathcal{H}$ with respect to the empirical measure over $X$, is defined as, $\|h_1 - h_2\|_{p,X} = \left( \frac{1}{m} \sum_{x \in X} |h_1(x) - h_2(x)|^p \right)^{1/p}$. Similarly, the distance with respect

---

[4] The term semi-DP algorithm has also been used in some works.

to the population, is given by $\|h_1 - h_2\|_{p, \mathcal{D}_{\mathcal{X}}} = \left( \mathbb{E}_{x \sim \mathcal{D}_{\mathcal{X}}} |h_1(x) - h_2(x)|^p \right)^{1/p}$. The covering number of $\mathcal{H}$ at scale $\alpha > 0$ and given dataset $X$, denoted as $\mathcal{N}_p(\mathcal{H}, \alpha, X)$ is the size of the minimal set of hypothesis, $\tilde{\mathcal{H}}$, such that for any $h \in \mathcal{H}$ there exists $\tilde{h}$ with $\|h - \tilde{h}\|_{p,X} \leq \alpha$. We define $\mathcal{N}_p(\mathcal{H}, \alpha, m) = \sup_{X : |X| = m} \mathcal{N}_p(\mathcal{H}, \alpha, X)$, the covering number with respect to all datasets of size $m$. We define fat-shattering dimension below.

**Definition 2.** *[BLW94] Let $\mathcal{H} \subseteq \mathbb{R}^{\mathcal{X}}$ and $\alpha > 0$. We say that $\mathcal{H}$ $\alpha$-shatters $X = \{x_1, x_2, \ldots, x_m\}$ if $\sup_{r \in \mathbb{R}^m} \min_{y \in \{-1,1\}^m} \sup_{h \in \mathcal{H}} \min_{i \in [m]} y_i(h(x_i) - r_i) \geq \alpha$. The fat-shattering dimension, $\text{fat}_\alpha(\mathcal{H})$, is the size of the largest $\alpha$-shattered set.*

We define $\mathfrak{R}_m(\mathcal{H})$, the worst-case Rademacher complexity of $\mathcal{H}$ with respect to $m$ data points, as $\mathfrak{R}_m(\mathcal{H}) = \sup_{X : |X| = m} \mathbb{E}_{\sigma_i} \sup_{h \in \mathcal{H}} \frac{1}{m} \sum_{i=1}^m \sigma_i h(x_i)$. An important example is that of norm-bounded linear predictors $\mathcal{H} = \{w : x \mapsto \langle w, x \rangle : \|w\| \leq D\}$ over $\mathcal{X} = \{x : \|x\| \leq \|\mathcal{X}\|\}$. Herein, $\text{fat}_\alpha(\mathcal{H}) = \Theta\left( \frac{D^2 \|\mathcal{X}\|^2}{\alpha^2} \right)$ and $\mathfrak{R}_m(\mathcal{H}) = \Theta\left( \frac{D\|\mathcal{X}\|}{\sqrt{m}} \right)$ [KST08, SST10].

# 3 Private Stochastic Convex Optimization with Labeled Public Data

In this section, we present our lower bounds for private stochastic convex optimization with public data. When interpreting the following results, it is helpful to note that in the nontrivial regime, $n_{\text{pub}} = \Theta(n)$ and $n_{\text{pub}} = o(n)$, although our results hold regardless. Further, recall that an upper bound for this problem of $O\left( R \min \left\{ \frac{1}{\sqrt{n_{\text{pub}}}}, \frac{1}{\sqrt{n}} + \frac{\sqrt{d \log(1/\delta)}}{n\epsilon} \right\} \right)$ can be obtained by simply either applying an optimal SCO algorithm to only the public data (and discarding the private data) or applying an optimal DP-SCO algorithm and treating the entire dataset as private [BFTGT19]. As we will see, this strategy is essentially optimal.

## 3.1 Lower Bound for Stochastic Convex Optimization

We start by stating our lower bound for public-data assisted differentially private SCO.

**Theorem 1.** *Let $\delta \leq \frac{1}{16nd}$, $\epsilon \leq 1$, and $d$ be larger than some universal constant. For any $(\epsilon, \delta)$-PA-DP algorithm, there exists a distribution $\mathcal{D}$, and a $G$-Lipschitz loss such that $\mathbb{E}\left[ L(\mathcal{A}(S_{pub}, S_{priv}); \mathcal{D}) - \min_{w : \|w\| \leq D} \{L(w; \mathcal{D})\} \right] = \Omega\left( GD \cdot \Psi(n_{pub}, n, d, \epsilon, \delta) \right)$, where for some universal constant $c$,*

$$
\Psi(n_{pub}, n, d, \epsilon, \delta) = \begin{cases} \min\left\{ \frac{1}{\sqrt{n_{pub}}}, \frac{1}{\sqrt{n}} + \frac{\sqrt{d \log(1/\delta)}}{n\epsilon} \right\}, & d \geq cn\epsilon, \; n_{pub} \leq \frac{n\epsilon}{c \log\left( 1/[\sqrt{nd}\delta] \right)} \\ \min\left\{ \frac{1}{\sqrt{n_{pub}}}, \frac{1}{\sqrt{n}} + \frac{\sqrt{d}}{n\epsilon} \right\}, & else \end{cases}
$$

The function $\Psi$ is defined to avoid repetitive notation in the rest of this section. Barring the mild restriction on $n_{\text{pub}}$, even though the $\sqrt{\log(1/\delta)}$ term is only obtained when $d \geq n\epsilon$, the "aggregate" lower bound is tight for all $d \notin [\frac{n\epsilon^2}{\log(1/\delta)}, n\epsilon]$ since when $d \leq \frac{n\epsilon^2}{\log(1/\delta)}$ the non-private $\frac{1}{\sqrt{n}}$ lower bound dominates. It is also pertinent to our results in Section 4 that the problem construction used to achieve this lower bound is a convex GLM, and as a result this lower bound holds even for GLMs.

Finally, similar statements can be made about strongly convex optimization. We again provide just one such statement here.

**Theorem 2.** *Let $\delta \leq \frac{1}{16nd}$, $\epsilon \leq 1$. For any $(\epsilon, \delta)$-PA-DP algorithm there exists a distribution $\mathcal{D}$, $\lambda$-strongly convex and $G$-Lipschitz loss such that*

$$
\mathbb{E}\left[ L(\mathcal{A}(S_{pub}, S_{priv}); \mathcal{D}) - \min_{w : \|w\| \leq D} \{L(w; \mathcal{D})\} \right] = \Omega\left( \frac{G^2}{\lambda} \Psi^2(n_{pub}, n, \epsilon, \delta) \right).
$$

The crux of the proofs for both the above results lies in establishing new mean estimation lower bounds for PA-DP mean estimation, which we give in Appendix B.1. These mean estimation lower bound use a novel application of a construction known as fingerprinting codes. In particular, the introduction of public data introduces significant challenges in the traditional analysis of fingerprinting codes. As these challenges are more technical in nature, we defer their discussion to Appendix B.2.

After establishing the mean estimation lower bounds, we can adapt the reductions first used in [BST14] that show mean estimation lower bounds can be used to provide lower bounds for risk minimization without public data. Full proofs for the above claims, and in particular details for the above reductions, are found in Appendix B.3.

**Lower Bound for Pure DP Case.** While not the primary focus of this work, the previous lower bound directly leads to a lower bound for pure PA-DP SCO. Since any $\epsilon$-DP algorithm is $(\epsilon, \delta)$-DP for any $\delta > 0$, we can use the above theorem to obtain a non-trivial lower bound for the pure DP case by setting $\delta$ small. Specifically, by setting $\delta$ such that $\log(1/\delta) = \frac{n\epsilon}{120^2 n_{\text{pub}}}$, one immediately obtains a lower bound of $\Omega\left(\min\{\frac{1}{\sqrt{n_{\text{pub}}}}, \frac{\sqrt{d}}{\sqrt{n_{\text{pub}} \cdot n\epsilon}}\}\right)$ for $d$ large enough. Simplifying this expression yields the following.

**Corollary 1.** *Let $d \geq cn\epsilon$ for a constant $c$, and $\mathcal{A}$ be an $\epsilon$-PA-DP algorithm. There exist a distribution $\mathcal{D}$ and a $G$-Lipschitz loss such that $\mathbb{E}\left[L(\mathcal{A}(S_{pub}, S_{priv}); \mathcal{D}) - \min\limits_{w:\|w\| \leq D}\{L(w; \mathcal{D})\}\right] = \Omega\left(\frac{GD}{\sqrt{n_{pub}}}\right)$.*

The known $O\left(GD \min\left\{\frac{1}{\sqrt{n_{\text{pub}}}}, \frac{1}{\sqrt{n}} + \frac{d}{n\epsilon}\right\}\right)$ upper bound for this problem shows that this bound is tight (in the regime in which it holds). Essentially, this bound states that when $d \geq n\epsilon$, the public dataset is not useful (at least asymptotically). Previously [LLHR23, Theorem 31] established that when $d \leq n\epsilon/n_{\text{pub}}$, a tight lower bound of $\Omega\left(GD\left(\frac{d}{n\epsilon} + \frac{1}{\sqrt{n}}\right)\right)$ holds, effectively showing that in this regime the public dataset is not useful[5]. We leave the remaining regime where $d \in \left(\frac{n\epsilon}{n_{\text{pub}}}, n\epsilon\right)$ as an interesting open problem for future work. Finally, we note that similar statements can be made about strongly convex losses using Theorem 2.

# 4 Private Supervised Learning with Unlabeled Public Data

In this section, we consider supervised learning with real-valued predictors given labeled private data and unlabeled public data. Our results show that, in this setting, it is possible to go beyond the limitations established in the prior section.

## 4.1 Efficient PA-DP learning of Convex Generalized Linear Models

We start with learning linear predictors with convex loses a.k.a. convex generalized learning models. We propose Algorithm 1, which uses the public unlabeled data to perform dimensionality reduction of the private labeled feature vectors. In the following, we use span to denote the span of a set of vectors and dim to denote the dimension of a subspace. The dataset of public unlabeled feature vectors is denote as $X_{\text{pub}}$. Our algorithm projects the private feature vectors onto the subspace spanning $\mathcal{W} \cap \text{span}(X_{\text{pub}})$ to get $\dim(\text{span}(X_{\text{pub}}) \cap \mathcal{W})$-dimensional representation of the private feature vectors. It then reparametrizes the loss function so that its domain is $\dim(\text{span}(S_{\text{pub}}) \cap \mathcal{W})$-dimensional and applies a private subroutine in the lower dimensional space. The output of the subroutine is then embedded back in $\mathbb{R}^d$. Algorithms similar to Algorithm 1 have appeared in the literature (e.g. [PHYS24]). We emphasize that our key contribution is the formal analysis of this technique and the fact that we provide tight upper and lower bounds while simultaneously avoiding many of the strong assumptions seen in previous work, such as large margin assumptions.

---

**Algorithm 1** Efficient PA-DP learning of GLMs with unlabeled public data

---

**Input:** Private labeled dataset $S_{\text{priv}}$, public unlabeled dataset $X_{\text{pub}}$, privacy parameters $\epsilon, \delta > 0$.
  1: Let $U \in \mathbb{R}^{d \times \dim(\mathcal{W} \cap \text{span}(X_{\text{pub}}))}$ denote the orthogonal projection onto $\text{span}(X_{\text{pub}}) \cap \mathcal{W}$.
  2: Define $\tilde{S}_{\text{priv}} = \left\{(U^\top x_i, y_i)\right\}_{i=1}^{n_{\text{priv}}}$ and let $\tilde{\mathcal{W}} = \left\{U^\top w : w \in \mathcal{W}\right\}$.
  3: Apply $(\epsilon, \delta)$-DP subroutine, $\tilde{\mathcal{A}}$, on loss function $w \mapsto \phi_y(\langle w, x \rangle)$ with dataset $\tilde{S}_{\text{priv}}$ over the constraint set $\tilde{\mathcal{W}}$, to get $\tilde{w} \in \mathbb{R}^{\dim(\mathcal{W} \cap \text{span}(S_{\text{pub}}))}$.
**Output:** $\widehat{w} = U\tilde{w}$.

---

[5]This claim is based on a simplification of their theorem statement. Specifically, because $n_{\text{pub}} \leq n\epsilon/d$, which also implies $d \leq n\epsilon$, their lower bound $\Omega\left(R \min\left\{\frac{1}{\sqrt{n_{\text{pub}}}}, \frac{d}{n\epsilon} + \frac{1}{\sqrt{n}}\right\}\right)$ simplifies.

Our main result for convex Lipschitz losses is the following.

**Theorem 3.** *Let $\epsilon > 0, \delta > 0$ and $\epsilon \leq \log(1/\delta)$. For a G-Lipschitz, B-bounded convex loss function, Algorithm 1 satisfies $(\epsilon, \delta)$-PA-DP. If the private subroutine $\tilde{\mathcal{A}}$ guarantees the following, with probability at least $1 - \beta$,*

$$\widehat{L}(\tilde{\mathcal{A}}(\tilde{S}_{priv}); \tilde{S}_{priv}) - \min_{w \in \tilde{\mathcal{W}}} \widehat{L}(w; \tilde{S}_{priv}) = O\left(GD \left\|\mathcal{X}\right\| \left(\frac{\sqrt{n_{pub} \log(1/\delta)} + \sqrt{\log(1/\beta)}}{n_{priv}\epsilon}\right)\right) \quad (1)$$

*then with $n_{pub} = \tilde{O}\left(\frac{n_{priv}\epsilon}{(\log(2/\beta) + \log(1/\delta))^{1/2}}\right)$, with probability at least $1 - \beta$, $L(\widehat{w}; \mathcal{D}) - L(w^*; \mathcal{D})$ is*

$$O\left(GD \left\|\mathcal{X}\right\| \left(\frac{\sqrt{\log(4/\beta)}}{\sqrt{n_{priv}}} + \frac{(\log(2/\beta) + \log(1/\delta))^{1/4}}{\sqrt{n_{priv}\epsilon}}\right) + \frac{B\sqrt{\log(4/\beta)}}{\sqrt{n_{priv}}}\right).$$

We note that DP algorithms such as projected noisy SGD [BST14] and the regularized exponential mechanism [GLL22], both of which can be implemented efficiently, are can be used to achieve (1), since the projected problem is at most $n_{\text{pub}}$ dimensional.

The above result shows that in the usual regime of $\epsilon = \Theta(1)$, there is no *price* of privacy, thereby obtaining the non-private rate of $O\left(\frac{1}{\sqrt{n_{\text{priv}}}}\right)$. We contrast this with the rate of $O\left(\frac{1}{\sqrt{n_{\text{priv}}}} + \frac{\sqrt{d}}{n_{\text{priv}}\epsilon}\right)$, achievable without public data. Our result is better when $d \geq n_{\text{priv}}\epsilon$, which is the interesting regime since herein the private error dominates the non-private error. Further, our lower bound (Theorem 4 below) shows that this is the non-trivial regime (for any $\epsilon = O(1)$), since otherwise, even with unlimited public data, the optimal rate is achieved without using any of it. We also note that the above rate is achievable without public data, but in the unconstrained setting where the output $\widehat{w}$ can have very large norm and so may lie outside $\mathcal{W}$ [ABG$^+$22].

The proof of the result primarily follows from the more general result with fat-shattering hypothesis classes (Theorem 7). We provide the key ideas as well as some details pertaining to linear predictors in Section 4.2 after Theorem 7. The full proof of this result is deferred to Appendix C.

**Lower Bounds.** The above rate as well as the number of public samples used are nearly-optimal. The first claim is due to the following result, which gives a lower bound on excess risk of DP algorithms under full knowledge of the marginal distribution, for Lipschitz GLMs. As unlabeled public data can only reveal information about the marginal distribution, this shows that further unlabeled public samples cannot hope to improve the rate we give in Theorem 3.

**Theorem 4.** *Let $\epsilon \leq 1, \delta \leq \epsilon$ and $\mathcal{A}$ be an $(\epsilon, \delta)$-DP algorithm. There exists a G-Lipschitz convex GLM loss function, and joint distribution $\mathcal{D}$ such that given a dataset $S$ comprising $n$ i.i.d. samples from $\mathcal{D}$ and full knowledge of the marginal distribution $\mathcal{D}_\mathcal{X}$, we have the following:*
$\mathbb{E}_{\mathcal{A},S}\left[L(\mathcal{A}(S); \mathcal{D}) - \min_{w:\|w\| \leq D} L(w; \mathcal{D})\right] = \Omega\left(GD \left\|\mathcal{X}\right\| \left(\frac{1}{\sqrt{n}} + \min\left\{\frac{1}{\sqrt{n\epsilon}}, \frac{\sqrt{d}}{n\epsilon}\right\}\right)\right).$

We note that the bound with $\frac{\sqrt{d}}{n_{\text{priv}}\epsilon}$ can be achieved without using any public data via standard results [BFTGT19, ABG$^+$22]. This result is largely a corollary of [ABG$^+$22, Theorem 6]. We provide full details in Appendix C.3.1.

To establish optimality of public sample complexity, we give the following lower bound which shows that $\tilde{\Omega}(n_{\text{priv}}\epsilon)$ samples are necessary to achieve the above rate. See Appendix C.3.2 for proof.

**Theorem 5.** *Let $n_{priv}, n_{pub}, d \in \mathbb{N}, \epsilon \leq 1, \delta < \frac{1}{16dn}$ and $d = \omega(n_{priv}\epsilon)$. If there exists an $(\epsilon, \delta)$-PA-DP algorithm $\mathcal{A}$, which, for any G-Lispschitz convex GLM, achieves excess risk*
$\mathbb{E}\left[L(\mathcal{A}(X_{pub}, S_{priv}); \mathcal{D}) - \min_{w:\|w\| \leq D} L(w; \mathcal{D})\right] = O\left(GD \left\|\mathcal{X}\right\| \left(\frac{1}{\sqrt{n_{priv}}} + \frac{\sqrt{\log(1/\delta)}}{\sqrt{n_{priv}\epsilon}}\right)\right)$, *for $S_{priv} \sim \mathcal{D}^{n_{priv}}$ and $X_{pub} \sim \mathcal{D}_\mathcal{X}^{n_{pub}}$, then $n_{pub} = \Omega(\frac{n_{priv}\epsilon}{\log(1/\delta)})$.*

**Optimistic rates.** We now consider additional assumptions that the loss function is non-negative and $H$-smooth, such as in the case of linear regression where $\phi_y(a) = (a - y)^2$. This is a well-studied setting [SST10] especially since it allows for obtaining optimistic rates: those that interpolate between a slow worst-case rate and a faster rate under (near) realizability or interpolation conditions. The main result is the following.

**Theorem 6.** *Let $\epsilon > 0, \delta > 0$ and $\epsilon \leq \log(1/\delta)$. For a G-Lipschitz, B-bounded non-negative H-smooth loss function, Algorithm 1 satisfies $(\epsilon, \delta)$-PA-DP. If the private subroutine $\tilde{\mathcal{A}}$ guarantees Equation (1) with probability at least $1 - \beta$, then with $n_{pub} = \tilde{O}\left(\frac{(HD\|\mathcal{X}\|)^{2/3}(n_{priv}\epsilon)^{2/3}}{G^{2/3}(\log(1/\delta))^{1/3}} + \frac{\sqrt{H}n_{priv}\epsilon\sqrt{\widehat{L}(\widehat{w}^*;S_{priv})}}{G\sqrt{\log(1/\delta)}}\right)$, with probability at least $1 - \beta$,*

$$L(\widehat{w}; \mathcal{D}) - \widehat{L}(\widehat{w}^*; S_{priv}) = \tilde{O}\left(\left(\frac{\sqrt{H}D\|\mathcal{X}\|}{\sqrt{n_{priv}\epsilon}} + \sqrt{\frac{B}{n_{priv}}}\right)\sqrt{\widehat{L}(\widehat{w}^*;S_{priv})} + \frac{H^{1/4}D\|\mathcal{X}\|\sqrt{G}\widehat{L}(\widehat{w}^*;S_{priv})^{1/4}}{\sqrt{n_{priv}\epsilon}}\right)$$

$$+ \tilde{O}\left(\frac{GD\|\mathcal{X}\|}{n_{priv}\epsilon} + \left(\frac{\sqrt{H}D^2\|\mathcal{X}\|^2 G}{n_{priv}\epsilon}\right)^{2/3} + \frac{H\|\mathcal{X}\|^2 D^2}{n_{priv}\epsilon} + \frac{B}{n_{priv}}\right)$$

*where $\widehat{w}^*$ is the minimizer of $\widehat{L}$ w.r.t $S_{priv}$ and $\tilde{O}$ hides $poly(\log(1/\delta), \log(1/\beta))$ terms.*

A similar result as above can be obtained with $\widehat{L}(\widehat{w}^*; S_{\text{priv}})$ replaced by $L(w^*; \mathcal{D})$ above – see Theorem 14 for the full theorem statement. This rate, in the worst-case, is essentially the same as that of Theorem 3, which is $\tilde{O}\left(\frac{1}{\sqrt{n_{\text{priv}}}} + \frac{1}{\sqrt{n_{\text{priv}}\epsilon}}\right)$. However, optimistically, when $L(w^*; \mathcal{D})$ or $\widehat{L}(\widehat{w}^*; S_{\text{priv}})$ is small, we get a faster rate of $\tilde{O}\left(\frac{1}{n_{\text{priv}}} + \frac{1}{(n_{\text{priv}}\epsilon)^{2/3}}\right)$. We note that this is seemingly weaker than what is known in the unconstrained setting, where [ABG+22] obtained a worst-case rate of $\tilde{O}\left(\frac{1}{\sqrt{n_{\text{priv}}}} + \frac{1}{(n_{\text{priv}}\epsilon)^{2/3}}\right)$. We show that we can recover this faster rate under an extra assumption that the global minimizer of the risk, lies in the constraint set $\mathcal{W}$ – note that this is trivially true in the unconstrained setup; see Theorem 15 for the statement.

We note that projected noisy SGD [BST14] and the regularized exponential mechanism [GLL22], both of which can be implemented efficiently, are possible choices for the private sub-routine $\tilde{\mathcal{A}}$ that realize the above theorem statements.

## 4.2 PA-DP Supervised learning of Fat-Shattering Classes

In this section, we consider a general supervised learning setting with fat-shattering hypothesis classes and potentially non-convex losses, with unlabeled public data. Our proposed algorithm is similar to that of [ABM19], which uses the pubic unlabeled data to construct a small finite, yet representative, subset of the hypothesis class. Our construction uses a cover of the hypothesis class with respect to the $\ell_2$ distance of predictions on the public data points. We then use the exponential mechanism to privately select a hypothesis using the empirical loss on private data as the score function.

We note that we operate under the pure DP setting (as opposed to approximate DP). Our techniques are based on selection which do not exhibit improved guarantees under approximate DP. Further, we note that, without public data, with non-convex losses, there is no separation of optimal rates between pure and approximate DP [GTU23].

---

**Algorithm 2** Supervised private learning with public unlabeled data

---

**Input:** Datasets $X_{\text{pub}}$ and $S_{\text{priv}}$, privacy parameter $\epsilon > 0$, scale of cover $\alpha > 0, \gamma > 0$.

1: Construct $\tilde{\mathcal{H}}$, a minimal $\alpha$-cover of $\mathcal{H}$, with respect to the following metric

$$\|h_1 - h_2\|_{2, X_{\text{pub}}} = \sqrt{\frac{1}{n_{\text{pub}}}\sum_{x \in X_{\text{pub}}}(h_1(x) - h_2(x))^2}$$

2: Return $\widehat{h}$ sampled with probability $p(h) \propto \exp\left(-\gamma\widehat{L}(h; S_{\text{priv}})\right)$ over $h \in \tilde{\mathcal{H}}$

---

Our main result for the Lispchitz setting is the following.

**Theorem 7.** *Algorithm 2 with $\gamma = \frac{2\min(B, GR)}{n_{priv}\epsilon}$ satisfies $\epsilon$-PA-DP. For any $\alpha > 0$ and $n_{pub} = O\left(\max\left(\frac{R^2\log(2/\beta)}{\alpha^2}, \min\left\{m : log^3(m)\mathfrak{R}_m^2(\mathcal{H}) \leq \alpha^2\right\}\right)\right) < \infty$, with probability at least $1 - \beta$, we have $L(\widehat{h}; \mathcal{D}) - \min_{h \in \mathcal{H}} L(h; \mathcal{D})$ is at most*

$$2G\mathfrak{R}_{n_{priv}}(\mathcal{H}) + O\left(\frac{B\sqrt{\log(4/\beta)}}{\sqrt{n_{priv}}}\right) + \tilde{O}\left(\frac{\min(B, GR)(fat_{c\alpha}(\mathcal{H}) + \log(4/\beta))}{n_{priv}\epsilon}\right) + 2G\alpha,$$

*where $c$ is an absolute constant.*

Our result shows that the model of PA-DP with unlabeled public data allows for obtaining nontrivial rates for supervised learning with any fat-shattering class, as is the case in the non-private setting. Further, in many standard settings, such as that of (Euclidean) GLMs, the Rademacher complexity is $\mathfrak{R}_m(\mathcal{H}) = O\left(\frac{1}{\sqrt{m}}\right)$ which implies that $\text{fat}_\alpha(\mathcal{H}) = O\left(\frac{1}{\alpha^2}\right)$ (see Theorem 9). In those cases, our guarantee simplifies to essentially yield a rate of $O\left(\mathfrak{R}_{n_{priv}}(\mathcal{H}) + \frac{1}{(n_{priv}\epsilon)^{1/3}} + \frac{1}{\sqrt{n_{priv}}}\right)$ – see Corollary 4 for the exact statement for GLMs.

**Proof Idea.** We briefly discuss some main ideas in the proof. The key is to show that if $\tilde{\mathcal{H}}$ is a cover of $\mathcal{H}$ with respect to the *empirical distance* on public feature vectors, $\|\cdot\|_{2,X_{\text{pub}}}$, then with enough public feature vectors, it is also a cover with respect to the *population distance* $\|\cdot\|_{2,\mathcal{D}_\mathcal{X}}$. This is captured in the following result.

**Lemma 1.** *Let $\tilde{\mathcal{H}}$ be a $\tau$-cover of $\mathcal{H}$ with respect to $\|\cdot\|_{2,X_{pub}}$. For $n_{pub} = O\left(\max\left(\frac{R^2\log(1/\beta)}{\alpha^2}, \min\left\{m : log^3(m)\mathfrak{R}_m^2(\mathcal{H}) \le \alpha^2\right\}\right)\right) < \infty$, for every $h \in \mathcal{H}$, with probability at least $1 - \beta$, there exists $\tilde{h} \in \tilde{\mathcal{H}}$ such that $\|h - \tilde{h}\|_{2,\mathcal{D}_\mathcal{X}} \le \alpha + \tau$.*

This result allows us to appropriately approximate a hypothesis class with enough public unlabeled points. This approximation roughly translates to the same additive error in the final bound while concurrently allowing for the use of the smaller finite hypothesis class $\tilde{\mathcal{H}}$ of size $|\tilde{\mathcal{H}}| = \tilde{O}(\text{fat}_\tau(\mathcal{H}))$.

For linear predictors with convex losses, as in Theorem 3, we show that the span$(X_{\text{pub}}) \cap \mathcal{W}$ is a valid 0-cover w.r.t. $\|\cdot\|_{2,X_{\text{pub}}}$. However, the cover being continuous and convex allows application of convex optimization techniques (as opposed to selection, as above), thereby obtaining stronger results with efficient procedures. The above procedure yields optimistic rates for non-negative and smooth losses; see Theorem 17 for details.

### 4.2.1 Application: Neural Networks

In this section, we instantiate our general result to give a guarantee for learning feed-forward neural networks in the PA-DP setting. We use the result of [GRS18] but note that other results which give bounds on the Rademacher complexity of neural networks, such as [BFT17, Sel23] can also be used.

We consider a depth $M$ feed-forward neural network which implements the function $x \mapsto W_M(\sigma(W_{M-1}\ldots\sigma(W_1x))\ldots)$. Here, $W_1, W_2, \ldots, W_M$ are the weight matrices and $\sigma$ is a (non-linear) activation function. We consider 1-Lipschtiz positive-homogeneous activation such as the ReLU function, $\sigma(z) = \max(0, z)$, applied coordinate-wise. Our main result is the following.

**Corollary 2.** *Let $(R_j)_{j=1}^M$ be a sequence of scalars and $M \in \mathbb{N}$. In the setting of Theorem 7 with $\mathcal{X} = \{x \in \mathbb{R}^d : \|x\| \le \|\mathcal{X}\|\}$ and $\mathcal{H}$ being the class of depth $M$ feed-forward neural networks, with 1-Lipschtiz positive-homogenous activation, and weight matrices, bounded as $\|W_j\|_F \le R_j$, with $n_{pub} = \tilde{O}\left((\|\mathcal{X}\|(\prod_{j=1}^M R_j))^{2/3}(n_{priv}\epsilon)^{2/3}M^{1/3}\log(2/\beta)\right)$, with probability at least $1 - \beta$, $L(\hat{h};\mathcal{D}) - \min_{h\in\mathcal{H}} L(h;\mathcal{D})$ is at most*

$$O\left(\frac{G\|\mathcal{X}\|\sqrt{M}\prod_{j=1}^M R_j}{\sqrt{n_{priv}}} + \frac{B\sqrt{\log(4/\beta)}}{\sqrt{n_{priv}}} + \frac{B\log(4/\beta)}{n_{priv}\epsilon}\right) + \tilde{O}\left(\left(\frac{BG^2M\|\mathcal{X}\|^2(\prod_{j=1}^M R_j)^2}{n_{priv}\epsilon}\right)^{1/3}\right).$$

We note that the above result has a polynomial dependence on the depth $M$, which is a consequence of the (non-private) Rademacher complexity of [GRS18]. It is also possible to get fully size-independent bounds by utilizing such existing results, however they require more stringent norm bounds on the weight matrices [Sel23]. Further, a similar result follows for non-negative smooth losses from [SST10], but we omit this extension for brevity.

### 4.2.2 Application: Non-Euclidean GLMs

In the non-Euclidean GLM setting, we consider $(\mathbb{X}, \|\cdot\|)$ as a $d$ dimensional (where $d \in \mathbb{N} \cup \{\infty\}$) Banach space, and $(\mathbb{W}, \|\cdot\|_*)$ is its dual space. The feature vectors $x$ are bounded as $\mathcal{X} = \{x \in \mathbb{X} : \|x\| \le \|\mathcal{X}\|\}$ and $\mathcal{W} \subseteq \{w \in \mathbb{W} : \Delta(w) \le D^r\}$ where $\Delta$ is a $r$-uniformly convex function[6] with respect to $\|\cdot\|_*$. A canonical example is the $(\ell_p, \ell_q)$-setup [KST08, FGV17], wherein the functions $\Delta(w) = \frac{\log(d)}{2} \|w\|_{1+(1/\log(d))}^2$, $\Delta(w) = \frac{1}{2(p-1)} \|w\|_p^2$ and $\Delta(w) = \frac{2^{p-2}}{p} \|w\|_p^p$ are 2, 2 and $p$-uniformly convex with respect to $\|\cdot\|_p$ for $p = 1$, $1 < p \le 2$ and $p \ge 2$ respectively.

The GLM loss function $\ell(w; x, y) = \phi_y(\langle w, x \rangle)$ where $\langle \cdot, \cdot \rangle : \mathcal{X} \times \mathcal{W} \to \mathbb{R}$ is a duality pairing. In this case, the Rademacher complexity of linear functions, is bounded as $O\left(\frac{D\|\mathcal{X}\|}{m^{1/r}}\right)$, where $s$ is the conjugate of $r$ i.e. $\frac{1}{r} + \frac{1}{s} = 1$ (see, e.g. [FGV17]). We obtain the following result by instantiating Theorem 7 with the Rademacher complexity and fat-shattering dimension of non-Euclidean GLMs.

**Corollary 3.** *In the setting of Theorem 7, together with* $\mathcal{X} = \{x \in \mathbb{R}^d : \|x\| \le \|\mathcal{X}\|\}$ *and* $\mathcal{H} = \{x \mapsto \langle w, x \rangle, x \in \mathcal{X}, \Delta(w) \le D^r\}$. *Given* $n_{pub} = \tilde{O}\left((n_{priv}\epsilon)^{r/(r+1)} \log(2/\beta)\right)$, *with probability at least* $1 - \beta$, $L(\widehat{w}; \mathcal{D}) - \min_{w \in \mathcal{W}} L(w; \mathcal{D})$ *is at most*

$$\tilde{O}\left(GD\|\mathcal{X}\| \left(\frac{1}{n_{priv}^{1/r}} + \frac{\sqrt{\log(4/\beta)}}{\sqrt{n_{priv}}} + \frac{\log(2/\beta)}{(n_{priv}\epsilon)^{\frac{1}{r+1}}} + \frac{\log(4/\beta)}{n_{priv}\epsilon}\right) + \frac{B\sqrt{\log(4/\beta)}}{\sqrt{n_{priv}}}\right).$$

The above yields guarantees for the special case of $(\ell_p, \ell_q)$-setup with $r = \max\{2, p\}$. We remind that in the (constrained) *convex* Euclidean GLM setting, our dimension-independent rate in Theorem 3, with public unlabeled data recover the rates which were known to be achievable in the unconstrained setting. Further the above rate for $p = 1$ case can be used to obtain guarantees for the polyhedral setting with $\|w\|_1 \le D$ constraint, resulting in a $O\left(\sqrt{\frac{\log(d)}{n_{priv}}} + \left(\frac{\log(d)}{n_{priv}\epsilon}\right)^{1/3}\right)$ rate. We note that [BGM21] showed a rate of $\tilde{O}\left(\sqrt{\frac{\log(d)}{n}} + \frac{\sqrt{\log(d)}}{\sqrt{n\epsilon}}\right)$ for this setting, with convex losses without public data. Importantly, for the other cases, i.e. $p > 1, p \ne 2$, there are no such (nearly) dimension-independent analogs of our result without public data, as of yet.

## Acknowledgments and Disclosure of Funding

R. Arora's and E. Ullah's research was supported, in part, by NSF BIGDATA award IIS-1838139 and NSF CAREER award IIS-1943251. R. Bassily's and M. Menart's research was supported by NSF CAREER Award 2144532 and, in part, by NSF Award 2112471. C. Guzmán's research was partially supported by INRIA Associate Teams project, ANID FONDECYT 1210362 grant, ANID Anillo ACT210005 grant, and National Center for Artificial Intelligence CENIA FB210017, Basal ANID.

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

## A  Additional Preliminaries

**Theorem 8.** *[RS12, Theorem 12.7] For $\mathcal{H} \subseteq [-R, R]^{\mathcal{X}}$, $m \in \mathbb{N}$, $p \geq 1$, $0 < \alpha \leq R$, we have that*

$$\mathcal{N}_2(\mathcal{H}, \alpha, m) \leq \left(\frac{2R}{\alpha}\right)^{Cfat_{c\alpha}(\mathcal{H})}.$$

*Further, for any $\tau \in (0, 1)$,*

$$\log\left(\mathcal{N}_\infty(\mathcal{H}, \alpha, m)\right) \leq C'fat(\mathcal{H}, c'\tau\alpha) \log\left(\frac{Rm}{fat(\mathcal{H}, c'\tau\alpha)\alpha}\right) log^\tau\left(\frac{m}{fat(\mathcal{H}, c'\tau\alpha)}\right),$$

*where $c, c', C$ and $C'$ are absolute constants.*

**Theorem 9.** *[SST10, Lemma A.3] For any hypothesis class $\mathcal{H}$, any sample size $m$ and any $\alpha > \mathfrak{R}_m(\mathcal{H})$, we have that,*

$$fat_\alpha(\mathcal{H}) \leq \frac{4m\mathfrak{R}_m(\mathcal{H})^2}{\alpha^2}.$$

**Theorem 10.** *[GRS18, Theorem 1] Let $M \in \mathbb{N}$ and $(R_j)_{j=1}^M$ be a sequence of scalars. The Rademacher complexity of the class of depth $M$ neural networks with 1-Lipschitz, positive-homogeneous activation function, $\mathcal{H}$, with weights $\|W_j\|_F \leq R_j$ is bounded as,*

$$\mathfrak{R}_m(\mathcal{H}) \leq \frac{\|\mathcal{X}\|\left(\sqrt{2\log(2)M} + 1\right)\prod_{j=1}^M R_j}{\sqrt{m}}.$$

*Here $\|\cdot\|_F$ denotes the Frobenius norm.*

**Lemma 2.** *[DSS$^+$15, Implied by Lemmas 5 and 14] Let $f : \{\pm 1\}^n \mapsto \mathbb{R}$ and define $g : [-1, 1] \to \mathbb{R}$ as $g(p) = \underset{S \sim \mathcal{D}_p^n}{\mathbb{E}}[f(S)]$, where $\mathcal{D}_p$ is as defined in Appendix B. Then for $a, b \in \mathbb{R}$, $b > a$, and $\mu \sim \mathsf{Unif}([a, b])$,*

$$\underset{\mu, S}{\mathbb{E}}\left[f(S) \cdot \sum_{x \in S}(x - \mu)]\right] = \underset{\mu}{\mathbb{E}}\left[g'(\mu)(1 - \mu^2)\right]$$

$$= 1 - \underset{\mu}{\mathbb{E}}\left[\mu^2\right] + (g(b) - b)(1 - b^2)\frac{1}{|b - a|} - (g(a) - a)(1 - a^2)\frac{1}{|b - a|} + 2\underset{\mu}{\mathbb{E}}\left[(g(\mu) - \mu)\mu\right].$$

**Lemma 3.** *[FS17, Lemma A.1] Fix $\mu, \epsilon, \delta, \Delta \in \mathbb{R}$. Let $X$ and $Y$ be random variables supported on $[\mu - \Delta, \mu + \Delta]$. Suppose that $X$ and $Y$ are $(\epsilon, \delta)$-indistinguishable, that is for any $\mathcal{E} \subseteq \mathbb{R}$, $e^{-\epsilon}\left(\mathbb{P}[X \in \mathcal{E}] - \delta\right) \leq \mathbb{P}[Y \in \mathcal{E}] \leq e^\epsilon\mathbb{P}[X \in \mathcal{E}] + \delta$. Then*

$$|\mathbb{E}[X] - \mathbb{E}[Y]| \leq (e^\epsilon - 1)\mathbb{E}[|X - \mu|] + 2\delta\Delta.$$

## B  Missing proofs from Section 3

### B.1  Lower Bounds for Mean Estimation

As stated previously, our SCO result follows primarily from new lower bound for PA-DP mean estimation. Here, we consider the setting where $\mathcal{D}$ is supported on the $\ell_2$ ball of radius $R > 0$. We define the mean of $\mathcal{D}$ as $\mu(D)$. Our lower bound for PA-DP mean estimation follows much the same form as our SCO bound.

**Theorem 11.** *Let $\delta \leq \frac{1}{16nd}$, $\epsilon \leq 1$. For any $(\epsilon, \delta)$-PA-DP algorithm, there exists a distribution $\mathcal{D}$ such that $\mathbb{E}[\|\mathcal{A}(S_{pub}, S_{priv}) - \mu(\mathcal{D})\|] = \Omega\left(R \cdot \Psi(n_{pub}, n, \epsilon, \delta)\right)$.*

We present the full proof momentarily and provide a more detailed discussion on the challenges of establishing this lower bound in Appendix B.2. We highlight key ideas here. As with many other lower bounds in differential privacy, we leverage a construct known as fingerprinting codes [BUV14, DSS$^+$15]. A key aspect of our analysis is showing that fingerprinting distributions can be

used to recover the optimal *non-private* lower bound for mean estimation. This allows us to create a problem which is "hard" both privately and non-privately. The analysis works by first showing that any sufficiently accurate algorithm must strongly correlate with the sampled datapoints. Next, we show upper bounds on how strongly the output of the algorithm correlates with the sampled dataset. The method for upper bounding this correlation varies depending on whether a given datapoint is considered public or private. Combining these upper and lower bounds on correlation yields the claimed result.

To obtain the $\sqrt{\log(1/\delta)}$ factor term in the lower bound, we use similar ideas to those in [SU15, CWZ21]. However, the introduction of public data leads to complications in prior methods. As such, we show that by analyzing the correlation of the coordinate wise clipping of the algorithms output, we are able to get bounds that appropriately scale with the accuracy.

**Proof of Theorem 11**    Before proceeding, we introduce the so-called fingerprinting distribution which will be the basis of our hard instance for mean estimation [BUV14, DSS$^+$15]. Towards this end, for any vector $\mu \in [-1, 1]^d$ we define $\mathcal{D}_\mu$ as the product distribution where, for any $j \in [d]$, a sample has its $j$'th coordinate as 1 with probability $(1 + \mu_j)/2$ and as $-1$ with probability $(1 - \mu_j)/2$. As shorthand, we denote $\frac{R}{\sqrt{d}}\mathcal{D}_\mu$ as the distribution which samples a vector from $\mathcal{D}_\mu$ and then scales it by $\frac{R}{\sqrt{d}}$. For notational convenience, for a set $\mathcal{E}$, we will also use $\mathsf{Unif}(\mathcal{E})$ to denote the uniform distribution over elements of the set.

The theorem follows from two theorems which have different restrictions on the problem parameters. In addition to the following two theorems, Theorem 11 incorporates the classic $\frac{R}{\sqrt{n}}$ statistical lower bound that holds even non-privately. The first theorem we present holds for a larger range of parameters but does not achieve the dependence on $\log(1/\delta)$.

**Theorem 12.** *Let $\epsilon > 0$, $\delta \leq \frac{1}{16n}$ and $\mathcal{A}$ be an $(\epsilon, \delta)$-PA-DP algorithm. For any setting of $\min\left\{\frac{\sqrt{d}}{n_{priv}\epsilon}, \frac{1}{\sqrt{n_{pub}}}\right\} \leq M \leq 1$, if $\mu \sim \mathsf{Unif}([-M, M]^d)$ and $(S_{pub}, S_{priv}) \sim \frac{R}{\sqrt{d}}\mathcal{D}_\mu^n$ it holds that*

$$\mathop{\mathbb{E}}_{\mathcal{A}, S, \mu}\left[\|\mathcal{A}(S_{pub}, S_{priv}) - \mu(\mathcal{D})\|\right] = \Omega\left(R\min\left\{\frac{1}{\sqrt{n_{pub}}}, \frac{\sqrt{d}}{n(e^\epsilon - 1)}\right\}\right).$$

In application to Theorem 11, we use $(e^\epsilon - 1) \leq 2\epsilon$ whenever $\epsilon \leq 1$. The second theorem requires $d \geq n\epsilon$ but has the benefit of scaling with $\log(1/\delta)$.

**Theorem 13.** *Let $\delta \leq \frac{1}{3dn}$, $\epsilon \leq 1$, $d \geq 120^2 n\epsilon$, and $n_{pub} \leq \frac{n\epsilon}{120^2 \log(1/[\sqrt{nd}\delta])}$, and $\mathcal{A}$ an $(\epsilon, \delta)$-PA-DP algorithm. Then there exists $M > 0$ such for $\mu \sim \mathsf{Unif}([-M, M]^d)$ and $(S_{priv}, S_{pub}) \sim \frac{R}{\sqrt{d}}\mathcal{D}_\mu^n$ it holds that*

$$\mathop{\mathbb{E}}_{\mathcal{A}, S, \mu}\left[\|\mathcal{A}(S_{pub}, S_{priv}) - \mu(\mathcal{D})\|\right] = \Omega\left(R\min\left\{\frac{1}{\sqrt{n_{pub}}}, \frac{\sqrt{d\log(1/\delta)}}{n\epsilon}\right\}\right).$$

A crucial part of the analysis is leveraging the so called fingerprinting lemma, which roughly states that any accurate algorithm given a dataset sampled from $\mathcal{D}_\mu$ must strongly correlate with vectors in the dataset. Particularly pertinent to our analysis is achieving such a correlation even when the components of the mean $\mu$ are much smaller than 1. Towards this end, we leverage the robust distribution framework of [DSS$^+$15] to achieve the following version of the fingerprinting lemma.

**Lemma 4** (Fingerprinting Lemma). *Let $M \in [0, 1]$ and $\mu$ be sampled uniformly from $[-M, M]^d$. Let $\mathcal{A}$ satisfy $\mathop{\mathbb{E}}_{S\sim\mathcal{D}_\mu^n}[\|\mathcal{A}(S) - \mu\|] \leq \alpha$ (for any $\mu \in [-1, 1]^d$). Then one has*

$$\mathop{\mathbb{E}}_{\mathcal{A}, S, \mu}\left[\sum_{i=1}^n \langle \mathcal{A}(S), x_i - \mu \rangle\right] \geq \frac{2d}{3} - \frac{\alpha\sqrt{d}}{M} - 2M\sqrt{d}\alpha.$$

*Proof.* In the following we treat $\mathcal{A}$ as a deterministic function and bound $\mathop{\mathbb{E}}_{S, \mu}[\sum_{i=1}^n \langle \mathcal{A}(S), x_i - \mu \rangle]$. This is sufficient to bound $\mathop{\mathbb{E}}_{\mathcal{A}, S, \mu}[\sum_{i=1}^n \langle \mathcal{A}(S), x_i - \mu \rangle]$ for randomized $\mathcal{A}$, since the analysis holds for

any function (i.e. the distribution does not depend on $\mathcal{A}$). Further, we start with the one dimensional case such that $\mu \in \mathbb{R}$. Define $g(\mu) = \mathop{\mathbb{E}}_{S \sim \mathcal{D}_\mu^n} [\mathcal{A}(S)]$. We start by applying results developed in [DSS$^+$15],

$$
\mathop{\mathbb{E}}_{S,\mu} \left[ \mathcal{A}(S) \sum_{i=1}^n (x_i - \mu) \right] \overset{(i)}{=} \mathop{\mathbb{E}}_\mu \left[ g'(\mu)(1 - \mu^2) \right]
$$

$$
\overset{(ii)}{\geq} 1 - \mathbb{E}\left[\mu^2\right] + 2\mathop{\mathbb{E}}_\mu\left[(g(\mu) - \mu)\mu\right] - \frac{|g(-M) + M| + |g(M) - M|}{2M}
$$

$$
\geq 2/3 + 2\mathop{\mathbb{E}}_\mu\left[(g(\mu) - \mu)\mu\right] - \frac{|g(-M) + M| + |g(M) - M|}{2M}.
$$

Above, $(i)$ comes from [DSS$^+$15, Lemma 5] and $(ii)$ comes from [DSS$^+$15, Lemma 14], which we have collectively restated in Lemma 2. We now have

$$
\mathop{\mathbb{E}}_{S,\mu} \left[ \mathcal{A}(S) \sum_{i=1}^n (x_i - \mu) \right] \geq 2/3 + \frac{|g(-M) + M| + |g(M) - M|}{2M} + 2\mathop{\mathbb{E}}_\mu\left[(g(\mu) - \mu)\mu\right]
$$

$$
\geq 2/3 + \frac{|g(-M) - M| + |g(M) - M|}{2M} - 2\mathop{\mathbb{E}}_\mu\left[|g(\mu) - \mu| \cdot |\mu|\right]
$$

$$
\geq 2/3 - \frac{|\mathbb{E}_{S \sim \mathcal{D}_{-M}}[\mathcal{A}(S)] + M| + |\mathbb{E}_{S \sim \mathcal{D}_M}[\mathcal{A}(S)] - M|}{2M}
$$

$$
- 2M\mathop{\mathbb{E}}_\mu\left[\left| \mathop{\mathbb{E}}_{S \sim \mathcal{D}_\mu}[\mathcal{A}(S)] - \mu \right|\right].
$$

Above we use the fact that $|\mu| \leq M$ and the definition of $g$.

We can now extend the above analysis to higher dimensions. For $\mu \in \mathbb{R}^d$, the above holds for each $\mu_j, j \in [d]$. For convenience define $\bar{M} = (M, \ldots, M) \in \mathbb{R}^d$. Summing over $d$ dimensions we have

$$
\mathop{\mathbb{E}}_{S,\mu}\left[\left\langle \mathcal{A}(S), \sum_{i=1}^n (x_i - \mu) \right\rangle\right]
$$

$$
\geq \frac{2d}{3} - \frac{1}{2M}\left\|\mathop{\mathbb{E}}_{S \sim \mathcal{D}_{-\bar{M}}}[\mathcal{A}(S)] + \bar{M}\right\|_1 - \frac{1}{2M}\left\|\mathop{\mathbb{E}}_{S \sim \mathcal{D}_{\bar{M}}}[\mathcal{A}(S)] - \bar{M}\right\|_1 - 2M\mathop{\mathbb{E}}_\mu\left[\left\|\mathop{\mathbb{E}}_{S \sim \mathcal{D}_\mu}[\mathcal{A}(S)] - \mu\right\|_1\right]
$$

$$
\geq \frac{2d}{3} - \frac{1}{2M}\mathop{\mathbb{E}}_{S \sim \mathcal{D}_{-\bar{M}}}\left[\|\mathcal{A}(S) + \bar{M}\|_1\right] - \frac{1}{2M}\mathop{\mathbb{E}}_{S \sim \mathcal{D}_{\bar{M}}}\left[\|\mathcal{A}(S) - \bar{M}\|_1\right] - 2M\mathop{\mathbb{E}}_{S,\mu}\left[\|\mathcal{A}(S) - \mu\|_1\right]
$$

$$
\geq \frac{2d}{3} - \frac{\sqrt{d}}{2M}\mathop{\mathbb{E}}_{S \sim \mathcal{D}_{-\bar{M}}}\left[\|\mathcal{A}(S) + \bar{M}\|_2\right] - \frac{\sqrt{d}}{2M}\mathop{\mathbb{E}}_{S \sim \mathcal{D}_{\bar{M}}}\left[\|\mathcal{A}(S) - \bar{M}\|_2\right] - 2\sqrt{d}M\mathop{\mathbb{E}}_{S,\mu}\left[\|\mathcal{A}(S) - \mu\|_2\right]
$$

$$
\geq \frac{2d}{3} - \frac{\alpha\sqrt{d}}{M} - 2M\sqrt{d}\alpha.
$$

This proves the claim. $\qquad\square$

We now turn towards proving Theorems 12 and 13. We start with the simpler proof of Theorem 12.

*Proof of Theorem 12.* For our proof we will use a dataset of vectors in $\{\pm 1\}^d$, and as such the $\ell_2$ bound on the data is $\sqrt{d}$. The final result will follow from rescaling by $\frac{R}{\sqrt{d}}$.

Let $S = \{x_1, x_2, \ldots x_n\} = (S_{\text{pub}}, S_{\text{priv}}) \sim \mathcal{D}_\mu^n$ be the concatenation of the public and private datasets. We also define $\alpha = \mathbb{E}\left[\|\mathcal{A}(S_{\text{pub}}, S_{\text{priv}}) - \mu\|\right]$ for notational convenience.

Define the following statistics,

$$
Z_i = \langle \mathcal{A}(S_{\text{pub}}, S_{\text{priv}}) - \mu, x_i - \mu \rangle
$$

$$
Z_i' = \langle \mathcal{A}(S_{\text{pub}}, S_{\sim i}) - \mu, x_i - \mu \rangle.
$$

where $S_{\sim i}$ is the dataset formed by replacing $i$-th data point of $S_{\text{priv}}$ with $x_i' \sim \mathcal{D}_\mu$. We have,

$$\mathop{\mathbb{E}}_{\mathcal{A},S,\mu}\Big[\sum_{i=1}^{n} Z_i\Big] = \mathbb{E}\Big[\sum_{i=1}^{n_{\text{pub}}} Z_i\Big] + \mathbb{E}\Big[\sum_{i=n_{\text{pub}}+1}^{n} Z_i\Big]. \tag{2}$$

The lower bound proceeds by providing upper and lower bounds on the above sum. We first have

$$\mathbb{E}\left[\left\langle \mathcal{A}(S_{\text{pub}}, S_{\text{priv}}) - \mu, \sum_{i=1}^{n_{\text{pub}}} (x_i - \mu)\right\rangle\right] \leq \sqrt{\mathbb{E}[\|\mathcal{A}(S_{\text{pub}}, S_{\text{priv}}) - \mu\|^2]\mathbb{E}\left\|\sum_{i=1}^{n_{\text{pub}}}(x_i - \mu)\right\|^2}$$

$$\leq \alpha\sqrt{dn}.$$

where the first inequality used Cauchy-Schwartz.

For the second term in Equation (2), we utilize differential privacy. Specifically, [FS17, Lemma A.1], restated in Lemma 3, gives that

$$\mathbb{E}\Big[\sum_{i=n_{\text{pub}}+1}^{n} Z_i\Big] \leq \sum_{i=n_{\text{pub}}+1}^{n} \left(\mathbb{E}[Z_i'] + 2(e^\epsilon - 1)\sqrt{\text{Var}(Z_i')} + 8\delta d\right)$$

$$\leq 4n_{\text{priv}}(e^\epsilon - 1)\alpha + 8n_{\text{priv}}\delta d.$$

Above we use that $\text{Var}(Z_i') \leq 4\alpha^2$ since $\|x_i - \mu\|_\infty \leq 4$. Plugging the above two in Equation (2) yields,

$$\mathbb{E}\Big[\sum_{i=1}^{n} Z_i\Big] \leq (4n_{\text{priv}}(e^\epsilon - 1)\alpha + 8n_{\text{priv}}\delta d) + \alpha\sqrt{dn_{\text{pub}}}.$$

We now use the fingerprinting lemma, Lemma 4, to lower bound the correlation. In this regard, note $\mathop{\mathbb{E}}_{S,\mu}[\langle \mu, \sum_{i=1}^{n}(x_i - \mu)\rangle] = 0$. Thus

$$\mathbb{E}\Big[\sum_{i=1}^{n} Z_i\Big] = \mathbb{E}\left[\left\langle \mathcal{A}(S_{\text{pub}}, S_{\text{priv}}), \sum_{i=1}^{n} x_i - \mu\right\rangle\right] \geq \frac{2d}{3} - \frac{\alpha\sqrt{d}}{2M} - 2M\sqrt{d}\alpha.$$

Plugging the obtained upper bound on the left hand side gives us,

$$4n_{\text{priv}}(e^\epsilon - 1)\alpha + 8n_{\text{priv}}\delta d + \alpha\sqrt{dn_{\text{pub}}} \geq \frac{2d}{3} - \frac{\alpha\sqrt{d}}{M} - 2M\sqrt{d}\alpha$$

$$\implies 4n_{\text{priv}}(e^\epsilon - 1)\alpha + \alpha\sqrt{dn_{\text{pub}}} \geq \frac{d}{6} - \frac{\alpha\sqrt{d}}{M} - 2M\sqrt{d}\alpha$$

$$\implies \alpha\left(4n_{\text{priv}}(e^\epsilon - 1) + \sqrt{dn_{\text{pub}}} + \frac{\sqrt{d}}{M} + 2M\sqrt{d}\right) \geq \frac{d}{6}$$

$$\implies \alpha \geq \frac{1}{24}\min\left\{\frac{d}{n_{\text{priv}}(e^\epsilon - 1)}, \frac{\sqrt{d}}{\sqrt{n_{\text{pub}}}}, M\sqrt{d}, \frac{\sqrt{d}}{M}\right\}$$

$$\implies \alpha \geq \frac{1}{24}\min\left\{\frac{d}{n_{\text{priv}}(e^\epsilon - 1)}, \frac{\sqrt{d}}{\sqrt{n_{\text{pub}}}}, M\sqrt{d}\right\}.$$

Above the first implication uses the assumption that $\delta \leq \frac{1}{16n_{\text{priv}}}$. The last implication uses the fact that $M\sqrt{d} \leq \frac{\sqrt{d}}{M}$ since $M \leq 1$. Rescaling by a $\frac{R}{\sqrt{d}}$ factor yields the bound

$$\mathop{\mathbb{E}}_{\mathcal{A},S,\mu}[\|\mathcal{A}(S_{\text{pub}}, S_{\text{priv}}) - \mu\|] \geq \frac{R}{24}\min\left\{\frac{\sqrt{d}}{n_{\text{priv}}(e^\epsilon - 1)}, \frac{1}{\sqrt{n_{\text{pub}}}}, M\right\}.$$

Observe that any setting of $M \geq \min\left\{\frac{\sqrt{d}}{n_{\text{priv}}\epsilon}, \frac{1}{\sqrt{n_{\text{pub}}}}\right\}$ realizes the bound claimed in the theorem statement. □

We now turn towards achieving a dependence on $\delta$ to prove Theorem 13. To do this, we leverage the the idea of filling a dataset with copies of each fingerprinting code seen in previous work [SU15, CWZ21]. However, in our case the introduction of public data makes this argument more delicate and leads to modified techniques for upper bounding the correlation statistics. See our discussion in Section B.2 for more details on why this is necessary.

*Proof of Theorem 13.* Let $\alpha = \mathbb{E}\left[\|\mathcal{A}(S_{\text{pub}}, S_{\text{priv}}) - \mu\|\right]$, $\alpha^* = \frac{1}{125} \min\left\{\frac{d\sqrt{\log(1/\delta)}}{n\epsilon}, \sqrt{\frac{d}{n_{\text{pub}}}}\right\}$ and assume by way of contradiction that $\alpha < \alpha^*$. Let $k = \frac{1}{3\epsilon} \log\left(1/[\sqrt{dn}\delta]\right)$. Let $m = \frac{n}{k}$. We set $M = \frac{4}{\sqrt{d}}(\alpha^* + \sqrt{\frac{d}{m}})$. Let $\mu \sim \text{Unif}([-M, M]^d)$, $S_z = \{z_1, ..., z_m\} \sim \mathcal{D}_\mu$. Sample $S_{\text{priv}}, S_{\text{pub}} \sim \text{Unif}(\{z_1, ..., z_m\})$ and denote the combined dataset as $S = \{x_1, \ldots, x_n\} = (S_{\text{pub}}, S_{\text{priv}})$. Note that as in the proof of Theorem 12, we are starting by showing a lower bound for the case where the data is drawn from $\mathcal{D}_\mu$ instead of $\frac{R}{\sqrt{d}}\mathcal{D}_\mu$, and will rescale at the end of the proof.

To prove our lower bound, we will provide upper and lower bounds on correlation statistics w.r.t. the intermediate dataset $S_z$. We will also introduce a clipping procedure which helps better control the upper bound on correlation. In this regard, for each $j \in [m]$ define $Z_j = \langle \lfloor \mathcal{A}(S_{\text{pub}}, S_{\text{priv}}) \rfloor_M, z_j - \mu \rangle$, where $\lfloor v \rfloor_M$ denotes the operation of clipping every element of $v$ to $[-M, M]$. In the following, we will provide upper and lower bounds on $\mathbb{E}\left[\sum_{j=1}^m Z_i\right]$ and use this to show that $\alpha \leq \alpha^*$ implies a contradiction.

**Lower Bound on Correlation** We now want to lower bound $\mathbb{E}\left[\sum_{j=1}^m Z_j\right]$. Towards this end, we can apply fingerprinting lemma, Lemma 4, to the algorithm which outputs the clipping. For $\widehat{\alpha} > 0$, if $\mathbb{E}_{\mathcal{A},S}\left[\|\lfloor \mathcal{A}(S_{\text{pub}}, S_{\text{priv}}) \rfloor_M - \mu\|\right] \leq \widehat{\alpha}$, then this yields,

$$\mathbb{E}_{\mathcal{A},S,\mu}\left[\sum_{j=1}^m Z_i\right] \geq \frac{2d}{3} - \frac{\widehat{\alpha}\sqrt{d}}{M} - 2M\sqrt{d}\widehat{\alpha}.$$

Now observe that

$$\mathbb{E}\left[\|\lfloor \mathcal{A}(S_{\text{pub}}, S_{\text{priv}}) \rfloor_M - \mu\|\right] \leq \mathbb{E}\left[\|\mathcal{A}(S_{\text{pub}}, S_{\text{priv}}) - \mu\|\right]$$

$$\leq \mathbb{E}\left[\left\|\mathcal{A}(S_{\text{pub}}, S_{\text{priv}}) - \frac{1}{m}\sum_{z \in S_z} z\right\|\right] + \mathbb{E}\left[\left\|\frac{1}{m}\sum_{z \in S_z} z - \mu\right\|\right]$$

$$\leq \alpha^* + \sqrt{\frac{d}{m}}$$

In the last step we use the assumed contradiction that $\alpha \leq \alpha^*$. Thus it suffices to set $\widehat{\alpha} = \alpha^* + \sqrt{\frac{d}{m}}$. Now by the setting $M = \frac{4}{\sqrt{d}}\widehat{\alpha}$ and $\widehat{\alpha} \leq \frac{\sqrt{d}}{12}$, Eqn. (3) implies

$$\mathbb{E}_{\mathcal{A},S,\mu}\left[\sum_{j=1}^m Z_i\right] \geq \frac{2d}{3} - \frac{d}{4} - 8\widehat{\alpha}^2 \geq \frac{d}{3}. \tag{3}$$

**Bounding the Number of Copies in the Dataset** We now turn towards the more involved process of upper bounding $\mathbb{E}_{S,\mu}\left[\sum_{i=1}^n Z_i\right]$. To do this however, it will be first helpful to show that no datapoint in $S_z$ is copied into $S$ too many times.

The first step is showing that no point is copied too many times into $S$. For $j \in [m]$, let $\mathcal{Z}_j = \{i \in [n] : x_i = z_j\}$. Observe

$$
\begin{aligned}
\mathbb{P}\left[\exists j \in [m] : |\mathcal{Z}_j| \geq (\tau+1)k\right] &\leq \sum_{j=1}^{m} \mathbb{P}\left[|\mathcal{Z}_j| \geq (\tau+1)k\right] \\
&\leq m \exp\left(-\frac{3\tau^2 n}{4m(1-1/n)}\right) \\
&\leq m \exp\left(-\frac{3\tau^2 \log(1/\delta)}{8\epsilon}\right).
\end{aligned}
$$

The second inequality follows from Bernstein's inequality for the sum of $n$ Bernoulli random variables with mean $1/m$ and the fact that $\mathbb{E}[|Z_j|] = \frac{n}{m} = k$. Set $\tau = \sqrt{\frac{8\epsilon \log(dm)}{3 \log(1/\delta)}}$ and note since $\epsilon \leq 1$ and $\log(dm) \leq \log(dn) \leq \log(1/\delta)$ (since $\delta \leq \frac{1}{dn}$), we have that $\tau \leq 2$. Thus, denoting $E$ as the event where no point in $S_z$ is copied into $S$ more than $3k$ times, we establish

$$
\mathbb{P}[E^c] = \mathbb{P}\left[\exists j \in [m] : |\mathcal{Z}_j| \geq 3k\right] \leq \frac{1}{d}. \tag{4}
$$

**Upper Bound on Correlation** Under our model, we assume that $\mathcal{A}$ must treat all data in $S_{\text{priv}}$ as private. We will in fact only need to use the privacy property for a subset of samples in $S_{\text{priv}}$ to prove the correlation upper bound. Let $\mathcal{I}_{\text{priv}} \subseteq [m]$ denote the set of indices s.t. $j \in \mathcal{I}_{\text{priv}}$ if every copy of $z_j$ sampled into the overall dataset is in the private dataset $S_{\text{priv}}$; that is $\mathcal{I}_{\text{priv}} = \{j : (\forall x \in S_{\text{pub}}) \, x \neq z_j\}$. Let $\mathcal{I}_{\text{pub}} = [m] \setminus \mathcal{I}_{\text{priv}}$. Observe that $\mathcal{I}_{\text{priv}}$ may contain indices for points in $S_z$ which are never sampled into $S$. We will see this does not affect our analysis.

We have

$$
\mathbb{E}\left[\sum_{j=1}^{m} Z_i\right] = \mathbb{E}\left[\sum_{j \in \mathcal{I}_{\text{priv}}} \langle \lfloor \mathcal{A}(S_{\text{pub}}, S_{\text{priv}}) \rfloor_M, z_j - \mu \rangle + \sum_{j \in \mathcal{I}_{\text{pub}}} \langle \lfloor \mathcal{A}(S_{\text{pub}}, S_{\text{priv}}) \rfloor_M, z_j - \mu \rangle\right].
$$

The first term on the RHS can be bounded using the privacy property of $\mathcal{A}$. For any fixed $j \in \mathcal{I}_{\text{priv}}$, let $S'_{\text{priv}}$ denote the dataset which replaces every instance of $z_j$ in $S_{\text{priv}}$ with a copied fresh sample from $\mathcal{D}_\mu$. By the above analysis, conditional on the event $E$, at most $3k$ such points need to be replaced conditional on the event $E$. Since $\mathcal{A}(S_{\text{pub}}, S'_{\text{priv}})$ is independent of $z_j$, by the Chernoff Hoeffding bound,

$$
\mathbb{P}\left[\langle \lfloor \mathcal{A}(S_{\text{pub}}, S'_{\text{priv}}) \rfloor_M, z_j - \mu \rangle \geq \tau \mid E\right] \leq \exp\left(-\frac{\tau^2}{8dM^2}\right). \tag{5}
$$

Since $\mathcal{A}$ satisfies $k$-group privacy with parameters $\widehat{\epsilon} \leq 3k\epsilon$ and $\widehat{\delta} = e^{3k\epsilon}\delta$, we have

$$
\mathbb{P}\left[\langle \lfloor \mathcal{A}(S_{\text{pub}}, S_{\text{priv}}) \rfloor_M, z_j - \mu \rangle \geq \tau \mid E\right] \leq \exp\left(\widehat{\epsilon} - \frac{\tau^2}{2dM^2}\right) + \widehat{\delta}.
$$

Setting $\tau = M\sqrt{d\log(1/\delta)}$, we obtain

$$
\begin{aligned}
\mathbb{E}[Z_j \mid E] &\leq \tau + 2d\mathbb{P}[Z_j \geq \tau \mid E] \\
&\leq M\sqrt{d\log(1/\delta)} + 2de^{\widehat{\epsilon}}\delta + \widehat{\delta} \\
&\leq M\sqrt{d\log(1/\delta)} + 3de^{3k\epsilon}\delta \leq 4M\sqrt{d\log(1/\delta)}.
\end{aligned}
$$

The last inequality comes from the setting of $k = \frac{1}{3\epsilon}\log\left(\frac{1}{\sqrt{dn}\delta}\right)$ and the fact that $M \geq \frac{1}{\sqrt{n}}$. Repeating this argument for each $j \in \mathcal{I}$ we get

$$
\mathbb{E}\left[\sum_{j \in \mathcal{I}_{\text{priv}}} Z_j\right] \leq \mathbb{E}\left[\sum_{j \in \mathcal{I}_{\text{priv}}} Z_j \,\Big|\, E\right]\mathbb{P}[E] + mMd\mathbb{P}[E^c] \leq 5mM\sqrt{d\log(1/\delta)}.
$$

The last inequality uses the bound established on each $\mathbb{E}\left[Z_j \mid E\right]$, $j \in \mathcal{I}_{\text{priv}}$, above and the bound on $\mathbb{P}\left[E^c\right]$ from Eqn. (4).

To bound the correlation over the remaining vectors, we have

$$\mathbb{E}\left[\sum_{j \in \mathcal{I}_{\text{pub}}} Z_j\right] \leq \sqrt{\mathbb{E}[\|\lfloor\mathcal{A}(S_{\text{pub}}, S_{\text{priv}})\rfloor_M\|^2]\mathbb{E}\left[\Big\|\sum_{j \in \mathcal{I}_{\text{pub}}}(z_j - \mu)\Big\|^2\right]} \leq 2Md\sqrt{n_{\text{pub}}}.$$

Above we have used the fact that $|\mathcal{I}_{\text{pub}}| \leq n_{\text{pub}}$ because $i \in \mathcal{I}_{\text{pub}}$ only if at least one copy of $z_i$ is sampled into $S_{\text{pub}}$. Combining the above we have

$$\mathbb{E}\left[\sum_{j=1}^{m} Z_i\right] \leq 5mM\sqrt{d\log(1/\delta)} + 5Md\sqrt{n_{\text{pub}}}.$$

**Combining Bounds:** The previously derived lower bound in Eqn. (3) establishes that $\mathbb{E}\left[\sum_{j \in \mathcal{I}_{\text{priv}}} Z_j + \sum_{j \in \mathcal{I}_{\text{pub}}} Z_j\right] \geq \frac{d}{3}$. Using the above derived upper bounds we have the following manipulations,

$$M\sqrt{d}\left(m\sqrt{\log(1/\delta)} + \sqrt{dn_{\text{pub}}}\right) \geq \frac{d}{15}$$

$$\Longleftrightarrow \quad (\alpha^* + \sqrt{d/m})\left(m\sqrt{\log(1/\delta)} + \sqrt{dn_{\text{pub}}}\right) \geq \frac{d}{60}$$

$$\Longleftrightarrow \quad m\alpha\sqrt{\log(1/\delta)} + \alpha^*\sqrt{dn_{\text{pub}}} \geq \frac{d}{60} - \sqrt{d\log(1/\delta)\,m} - d\sqrt{\frac{n_{\text{pub}}}{m}}.$$

The second line above uses that $M = \frac{4}{\sqrt{d}}(\alpha^* + \sqrt{\frac{d}{m}})$. Under the condition that $n_{\text{pub}} \leq \frac{m}{120^2} \equiv n_{\text{pub}} \leq \frac{3n\epsilon}{120^2 \log(1/[\sqrt{nd}\delta])}$, which is satisfied under by assumption in the theorem statement, we have

$$m\alpha\sqrt{\log(1/\delta)} + \alpha^*\sqrt{dn_{\text{pub}}} \geq \frac{d}{120} - \sqrt{d\log(1/\delta)\,m}.$$

Now applying the assumption $d \geq 120^2 n\epsilon \implies m \leq \frac{d}{120^2 \log(1/\delta)}$ we obtain

$$m\alpha^*\sqrt{\log(1/\delta)} + \alpha^*\sqrt{dn_{\text{pub}}} \geq \frac{d}{120}$$

$$\alpha^* \geq \frac{1}{120}\min\left\{\frac{d\sqrt{\log(1/\delta)}}{n\epsilon}, \sqrt{\frac{d}{n_{\text{pub}}}}\right\}.$$

This establishes a contradiction, and thus $\alpha \geq \alpha^* = \frac{1}{125}\min\left\{\frac{d\sqrt{\log(1/\delta)}}{n\epsilon}, \sqrt{\frac{d}{n_{\text{pub}}}}\right\}$. Rescaling by $\frac{R}{\sqrt{d}}$ then yields the claimed result. $\qquad\square$

## B.2 Discussion of Lower Bound Analysis

We here provide more details on why the particular lower bound techniques we present were chosen. Our aim for the following discussion is to elucidate some of the subtleties of leveraging the fingerprinting code framework when public data is present, with the hope that it will aid future work on the characterization of PA-DP problems.

One crucial challenge in developing the mean estimation lower bounds in Appendix B.1 is ensuring that the correlation sum, traditionally defined as $\mathbb{E}\left[\sum_{x \in S}\langle\mathcal{A}(S), x - \mu\rangle\right]$, scales with the accuracy, $\alpha$, of the algorithm. Previous work, such as [CWZ21], achieves this by setting the underlying distribution, $\mathcal{D}$, to be a mixture distribution which, for some $p = o(1)$, samples a $0$ vector with probability $(1-p)$ and samples from the non-trivial distribution, $\mathcal{D}_\mu$, with probability $p$. However, now the variance satisfies $\mathbb{E}_{x \sim \mathcal{D}}\left[\|x - \mathbb{E}[x]\|^2\right] \leq 2pR^2$ meaning that when public data is present it

holds that $\mathbb{E}\left[\|\frac{1}{n_{\text{pub}}}\sum_{x\in S_{\text{pub}}}x - \mathbb{E}_{x\sim\mathcal{D}}\left[x\right]\|\right] \leq \frac{2pR}{\sqrt{n_{\text{pub}}}} = o\left(\frac{R}{\sqrt{n_{\text{pub}}}}\right)$, and one cannot hope to achieve the

desired lower bound. Alternatively, by instead analyzing the sum $\mathbb{E}\left[\sum_{x\in S}\langle\mathcal{A}(S) - \mu, x - \mu\rangle\right]$, as seen for example in [KU20], we are able to avoid sampling from a mixture distribution. Further, by leveraging the flexibility of the strong distribution framework from [DSS$^+$15], we are able to still ensure $\|\mu(\mathcal{D})\| = o(1)$, as needed for the SCO reduction; see Section B.3. These techniques lead to the result in Theorem 12.

Unfortunately, with regards to obtaining the $\sqrt{\log(1/\delta)}$ improvement in Theorem 13, the property $\mathbb{E}\left[\|\mathcal{A}(S) - \mu\|\right] \leq \alpha$ does little to help establish the needed tail bound; see Eqn. (5). By clipping the components of $\mathcal{A}(S)$ to to the range $[-O(\alpha), O(\alpha)]$, we are able to able to obtain the desired concentration. Unfortunately, this clipping technique in combination with the intermediate distribution, $\mathsf{Unif}(S_z)$, leads to the restrictions that $d \geq n\epsilon$ and $n_{\text{pub}} \leq \frac{n}{\log(1/[nd\delta])}$. These restrictions occur because of the need for the "additional error" introduced by the intermediate distribution to be negligible. To see this, observe the intermediate distribution leads to $\mathbb{E}\left[\|\mathcal{A}(S) - \mu\|\right] \geq \frac{1}{\sqrt{m}}$ since $\mathcal{A}(S)$ depends on only $m$ vectors from $\mathcal{D}_\mu$, and the analysis in the proof of Theorem 12 (with $n_{\text{priv}} = 0$) shows us that even non-private algorithms cannot do better on this distribution. We remark that [CWZ21] avoids this issue, and hence the restriction on $d$ and $n_{\text{pub}}$, because of the fact that one only actually needs $\|\mathbb{E}\left[\mathcal{A}(S)\right] - \mu\| \leq \alpha$ for the fingerprinting lemma to hold, and $\|\mathbb{E}\left[\mathcal{A}(S)\right] - \mu\| \leq \mathbb{E}\left[\|\mathcal{A}(S) - \mu\|\right]$. However, after clipping it is possible that $\|\mathbb{E}\left[\lfloor\mathcal{A}(S)\rceil_M\right] - \mu\| \geq \|\mathbb{E}\left[\mathcal{A}(S)\right] - \mu\|$.

### B.3 Missing proofs from Section 3.1

*Proof of Theorem 1.* We use the instance in [BST14], $\ell(w; x) = G\langle w, x\rangle$ and $\mathcal{W} = \left\{x \in \mathbb{R}^d : \|x\| \leq 1\right\}$. By a standard rescaling argument, we only need to consider $G = D = 1$. We will consider the re-scaled data distribution used in Theorem 12, where $\{z_1, ..., z_n\} \overset{i.i.d.}{\sim} \mathcal{D}_\mu$ and the dataset $S$ has $x_j = \frac{1}{\sqrt{d}}z_j$ for $j \in n$. Here $\mu \sim \mathsf{Unif}([-M, M]^d)$ where $M$ will be chosen later.

First note by Lemma 5 we have that $\mathbb{P}\left[\left|\|\mu\| - \sqrt{\frac{2}{3}}M\right| \geq \frac{M}{256}\right] \leq \frac{1}{512}$ so long as $d$ is larger than some constant. Define this event as $E$ and $E'$ its complement. Thus we have

$$
\begin{aligned}
\mathbb{E}\left[L(\mathcal{A}(S); \mathcal{D}) - L(w^*; \mathcal{D})\right] &= \mathbb{E}\left[L(\mathcal{A}(S); \mathcal{D}) - L(w^*; \mathcal{D})|E\right]\mathbb{P}[E] \\
&\quad + \mathbb{E}\left[L(\mathcal{A}(S); \mathcal{D}) - L(w^*; \mathcal{D})|E'\right]\mathbb{P}[E'] \\
&\geq \frac{1}{2}\mathbb{E}\left[L(\mathcal{A}(S); \mathcal{D}) - L(w^*; \mathcal{D})|E\right].
\end{aligned}
$$

Thus it suffices to lower bound the conditional excess risk.

The optimal solution under the aforementioned loss is $w^* = -\frac{\mu}{\|\mu\|}$, since the constraint set is a ball of radius 1. We can see that

$$
\begin{aligned}
L(\mathcal{A}(S); \mathcal{D}) - L(w^*; \mathcal{D}) &= \langle\mathcal{A}(S), \mu\rangle - \left\langle-\frac{\mu}{\|\mu\|}, \mu\right\rangle \\
&= \|\mu\|\left(1 - \langle\mathcal{A}(S), w^*\rangle\right) \\
&= \|\mu\|\left(1 - \frac{1}{2}\|\mathcal{A}(S)\|^2 - \frac{1}{2}\|w^*\|^2 + \frac{1}{2}\|\mathcal{A}(S) - w^*\|^2\right) \\
&\geq \frac{1}{2}\|\mu\|\|\mathcal{A}(S) - w^*\|^2.
\end{aligned}
\tag{6}
$$

We will now lower bound $\|\mathcal{A}(S) - w^*\|$ by using the lower bound for mean estimation developed in Theorem 11. Let the mean estimate candidate is $\bar{\mu}(S) = \bar{\mu} = -\sqrt{\frac{2}{3}}M\mathcal{A}(S)$. Under the event $E$,

$$
\begin{aligned}
\|\bar{\mu} - \mu\|^2 &= \left\| -\sqrt{2/3}M\mathcal{A}(S) - \mu \right\|^2 \\
&= \left\| -\|\mu\|\mathcal{A}(S) - \mu + (\sqrt{2/3}M - \|\mu\|)\mathcal{A}(S) \right\|^2 \\
&\leq 2M^2\|\mathcal{A}(S) - w^*\|^2 + \frac{M^2}{50} \\
\implies \|\mathcal{A}(S) - w^*\|^2 &\geq \frac{\|\bar{\mu} - \mu\|^2}{2M^2} - \frac{1}{512}.
\end{aligned}
\tag{7}
$$

The above follows from the definition of $w^*$ and since the algorithm's output is considered in a ball of radius 1, so $\|\mathcal{A}(S)\| \leq 1$.

Combining the above inequalities (6) and (7) then taking expectation we have,

$$
\begin{aligned}
\mathbb{E}\left[L(\mathcal{A}(S);\mathcal{D}) - L(w^*;\mathcal{D})|E\right] &\geq \mathbb{E}\left[\frac{1}{4}\|\mu\|\left(\frac{\|\bar{\mu} - \mu\|^2}{M^2} - \frac{1}{512}\right)\Bigg|E\right] \\
&\geq \frac{M}{1024}\left(\frac{\mathbb{E}\left[\|\bar{\mu} - \mu\|^2|E\right]}{2M^2} - \frac{1}{512}\right).
\end{aligned}
\tag{8}
$$

To bound $\mathbb{E}\left[\|\bar{\mu} - \mu\|^2|E\right]$, observe

$$
\begin{aligned}
\mathbb{E}[\|\bar{\mu} - \mu\|^2] &= \mathbb{E}[\|\bar{\mu} - \mu\|^2\,|E]\mathbb{P}[E] + \mathbb{E}_{\mu,S}[\|\bar{\mu} - \mu\|^2\,|E']\mathbb{P}[E'] \\
&\leq \mathbb{E}[\|\bar{\mu} - \mu\|^2\,|E] + 4M^2\mathbb{P}[E'].
\end{aligned}
$$

Rearranging we have

$$
\mathbb{E}[\|\bar{\mu} - \mu\|^2\,|E] \geq \mathbb{E}[\|\bar{\mu} - \mu\|^2] - \frac{M^2}{128}.
\tag{9}
$$

We will finish the bound by applying either Theorem 12 or Theorem 13.

**Via Theorem 12:** Set $M = \min\left\{\frac{\sqrt{d}}{8n_{\mathrm{priv}}}, \frac{1}{\sqrt{n_{\mathrm{pub}}}}\right\}$. Under this setting of $M$, Theorem 12 implies that the lower bound on mean estimate distance satisfies $\mathbb{E}\left[\|\bar{\mu} - \mu\|\right] \geq \frac{M}{8}$, and thus $\mathbb{E}\left[\|\bar{\mu} - \mu\|^2\,|E\right] \geq \frac{M^2}{128}$ by Eqn. (9) above. Plugging into Eqn. (8) we have

$$
\mathbb{E}\left[L(\mathcal{A}(S);\mathcal{D}) - L(w^*;\mathcal{D})\right] = \Omega\left(M\right) = \Omega\left(\min\left\{\frac{\sqrt{d}}{n_{\mathrm{priv}}}, \frac{1}{\sqrt{n_{\mathrm{pub}}}}\right\}\right).
$$

**Via Theorem 13:** In Theorem 13, the setting of $M$ used is

$$
\begin{aligned}
M &= 4\left(\frac{1}{125}\min\left\{\frac{\sqrt{d\log(1/\delta)}}{n\epsilon}, \sqrt{\frac{1}{n_{\mathrm{pub}}}}\right\} + \sqrt{\frac{\log\left(1/[\sqrt{d}n\delta]\right)}{n\epsilon}}\right) \\
&\leq \frac{1}{30}\min\left\{\frac{\sqrt{d\log(1/\delta)}}{n\epsilon}, \sqrt{\frac{1}{n_{\mathrm{pub}}}}\right\}.
\end{aligned}
$$

The inequality holds under the conditions $d \geq 120^2 n\epsilon$ and $n_{\mathrm{pub}} \leq \frac{n}{120^2\log(1/[\sqrt{n}d\delta])}$. Thus we have under this setting of $M$ that $\mathbb{E}\left[\|\bar{\mu} - \mu\|\right] \geq \frac{M}{8}$. Applying Eqns. (9) and (8) as in the previous case we have (providing the above conditions on $d$ and $n_{\mathrm{pub}}$ hold)

$$
\mathbb{E}\left[L(\mathcal{A}(S);\mathcal{D}) - L(w^*;\mathcal{D})\right] = \Omega\left(M\right) = \Omega\left(\min\left\{\frac{\sqrt{d\log(1/\delta)}}{n\epsilon}, \sqrt{\frac{1}{n_{\mathrm{pub}}}}\right\}\right).
$$

$\square$

**Lemma 5.** *For $z \sim \text{Unif}([-1,1]^d)$, we have that $\|z\| \in \frac{\sqrt{2d}}{\sqrt{3}} \pm \frac{\sqrt{3 \ln(1/\gamma)}}{2}$, with probability at least $1 - \gamma$.*

*Proof.* This follows from standard concentration of norm results. We have that,

$$\mathbb{E}\|z\|^2 = d\mathbb{E}z_1^2 = \frac{2d}{3}.$$

As in [Ver18, proof of Theorem 3.1.1], we use the simple fact that $|x - 1| > \delta \implies |x^2 - 1| > \max(\delta, \delta^2)$ for any $x, \delta \geq 0$, to get,

$$
\begin{aligned}
\mathbb{P}\left(\left|\|z\| - \frac{\sqrt{2d}}{\sqrt{3}}\right| > \frac{\sqrt{2d}\delta}{\sqrt{3}}\right) &= \mathbb{P}\left(\left|\frac{\sqrt{3}\|z\|}{\sqrt{2d}} - 1\right| > \delta\right) \\
&= \mathbb{P}\left(\left|\frac{3\|z\|^2}{2d} - 1\right| > \max(\delta, \delta^2)\right) \\
&= \mathbb{P}\left(\left|\frac{1}{d}\sum_{i=1}^{d} z_i^2 - \frac{2}{3}\right| > \max\left((2/3)\delta, ((2/3)\delta)^2\right)\right).
\end{aligned}
$$

We substitute $\bar{\delta} = \frac{2\delta}{3}$ and apply Bernstein's inequality for i.i.d sub-exponential random variables $z_i^2$. Since, $z_i \in [-1, 1]$, the sub-exponential norm $\leq 1$. Applying Corollary 2.8.3 from [Ver18], we get that,

$$\mathbb{P}\left(\left|\frac{1}{d}\sum_{i=1}^{d} z_i^2 - \frac{2}{3}\right| > \max\left(\bar{\delta}, (\bar{\delta})^2\right)\right) \leq \exp\left(-2\bar{\delta}^2 d\right) = \exp\left(-8\delta^2 d/9\right).$$

This gives us that

$$\mathbb{P}\left(\left|\|z\| - \frac{\sqrt{2d}}{\sqrt{3}}\right| > \frac{\sqrt{2d}\delta}{\sqrt{3}}\right) \leq \exp\left(-8\delta^2 d/9\right).$$

Hence, with probability, at least $1 - \gamma$, we have that $\|z\| \in \frac{\sqrt{2d}}{\sqrt{3}} \pm \frac{\sqrt{3 \ln(1/\gamma)}}{2}$, which completes the proof. $\qquad\square$

*Proof of Theorem 2.* We use the squared loss instance as in [BST14, Section 5.2]; $\ell(w; z) = \frac{\lambda}{2}\|w - z\|^2$, with $\|z\| \leq \frac{G}{2\lambda}$. The loss is $G$-Lipschitz and $\lambda$ strongly convex on the domain of unit ball at zero of radius $\frac{G}{2\lambda}$. Given a datasets $S = \{z_1, z_2, \ldots, z_n\}$, the population risk minimizer is simply the population mean $\mu(\mathcal{D})$. Further, it is straightforward to verify that the excess population risk a re-scaling of the mean estimation error

$$\mathbb{E}[L(\mathcal{A}(S)) - \min_w L(w)] = \frac{\lambda}{2}\mathbb{E}\|\mathcal{A}(S) - \mu(\mathcal{D})\|^2.$$

Substituting the mean estimation lower bounds, Theorem 11, completes the proof. $\qquad\square$

## C  Missing Proofs from Section 4

### C.1  Proof of Theorem 3

Define the orthogonal projection matrix $P_{X_{\text{pub}}} = UU^\top$. Note that the feature vectors in $\tilde{S}_{\text{priv}} = \{(U^\top x_i, y_i)\}_{i=1}^{n_{\text{priv}}}$ are bounded. In particular $\|U^\top x\|^2 = x^\top(UU^\top)x = x^\top P_{X_{\text{pub}}} x \leq \|x\|^2$, since $P_{X_{\text{pub}}}$ is an orthogonal projection onto $\text{span}(\mathcal{W} \cap S_{\text{pub}})$. Further, since $\tilde{w} \in \tilde{\mathcal{W}}$, we have that there exists $\mathring{w} \in \mathcal{W}$ such that $\tilde{w} = U^\top \mathring{w}$. Finally, $\widehat{w} = U\tilde{w} = UU^\top \mathring{w} = P_{X_{\text{pub}}} \mathring{w} \in \mathcal{W}$ since the range of $P_{X_{\text{pub}}} \subseteq \mathcal{W}$.

The privacy guarantee follows from the privacy guarantee of sub-routine $\tilde{\mathcal{A}}$. For utility, we define $w^* \in \arg\min_{w \in \mathcal{W}} L(w; \mathcal{D})$ and $\tilde{w}^* \in \arg\min_{w \in \tilde{\mathcal{W}}} L(w; U^\top \mathcal{D})$, where $U^\top \mathcal{D}$ denotes the distribution which first samples from $\mathcal{D}$ then project using $U^\top$. Let $\mathring{w}^* \in \mathcal{W}$ such that $\mathring{w}^* = U\tilde{w}^*$.

Note that from the GLM structure, $L(\tilde{w}^*; U^\top \mathcal{D}) = L(\mathring{w}^*; \mathcal{D})$. We have,

$$
\begin{aligned}
L(\widehat{w}; \mathcal{D}) - L(w^*; \mathcal{D}) &= L(\widehat{w}; \mathcal{D}) - \widehat{L}(\widehat{w}; S_{\text{priv}}) + L(\mathring{w}^*; \mathcal{D}) - L(w^*; \mathcal{D}) \\
&\quad + \widehat{L}(\tilde{w}^*; S_{\text{priv}}) - L(\mathring{w}^*; \mathcal{D}) + \widehat{L}(\widehat{w}; S_{\text{priv}}) - \widehat{L}(\mathring{w}^*; S_{\text{priv}}) \\
&= O\left( G\mathfrak{R}_{n_{\text{priv}}}(\mathcal{H}) + \frac{B\sqrt{\log(4/\beta)}}{\sqrt{n_{\text{priv}}}} \right) \\
&\quad + L(\mathring{w}^*; \mathcal{D}) - L(w^*; \mathcal{D}) + \widehat{L}(\tilde{w}; \tilde{S}_{\text{priv}}) - \min_{w \in \tilde{\mathcal{W}}} \widehat{L}(w; \tilde{S}_{\text{priv}}) \qquad (10)
\end{aligned}
$$

with probability at least $1 - \beta/4$. In the above, we control the generalization gap via uniform convergence and concentration for the fixed $\mathring{w}^*$ with respect to $S_{\text{priv}}$.

The last term $\widehat{L}(\tilde{w}; \tilde{S}_{\text{priv}}) - \min_{w \in \tilde{\mathcal{W}}} \widehat{L}(w; \tilde{S}_{\text{priv}})$ is bounded by the guarantee of the private sub-routine with probability at least $1 - \beta/4$,

$$
\widehat{L}(\tilde{\mathcal{A}}(\tilde{S}_{\text{priv}}); \tilde{S}_{\text{priv}}) - \min_{w \in \tilde{\mathcal{W}}} \widehat{L}(\tilde{\mathcal{A}}(\tilde{S}_{\text{priv}}); \tilde{S}_{\text{priv}}) = \tilde{O}\left( GD\|\mathcal{X}\| \left( \frac{\sqrt{n_{\text{pub}} \log(1/\delta)} + \sqrt{\log(4/\beta)}}{n_{\text{priv}}\epsilon} \right) \right).
$$

Finally, for any $\bar{w}^*$ such that $\bar{w}^* \in U\tilde{\mathcal{W}}$, with probability at least $1 - \beta/2$, from $G$-Lipschitznes, we have

$$
L(\mathring{w}^*; \mathcal{D}) - L(w^*; \mathcal{D}) \leq L(\bar{w}^*; \mathcal{D}) - L(w^*; \mathcal{D}) \leq G\|\bar{w}^* - w^*\|_{2, \mathcal{D}_{\mathcal{X}}} \leq G\alpha,
$$

where the last inequality follows essentially from Lemma 6 and Lemma 1 for $n_{\text{pub}} = O\left( \max\left( \frac{R^2 \log(2/\beta)}{\alpha^2}, \min\left\{ m : \log^3(n_{\text{pub}})\mathfrak{R}_{n_{\text{pub}}}^2(\mathcal{H}) \leq \alpha^2 \right\} \right) \right)$. To elaborate, the first step holds since $\mathring{w}^* = U\tilde{w}^*$ and $\tilde{w}^*$ is the the minimizer of risk over $\tilde{\mathcal{W}}$. Now, Lemma 1 guarantees that for any $w^* \in \mathcal{W}$, there exists a $\bar{w}^*$ in its $\alpha$-cover with respect to $\|\cdot\|_{2, X_{\text{pub}}}$, with $\|w^* - \bar{w}^*\|_{2, \mathcal{D}_{\mathcal{X}}} \leq \alpha$. To argue why $\text{span}(X_{\text{pub}})$ is an $\alpha$-cover, from Lemma 6, we have that from any $\alpha$-cover $\bar{\mathcal{W}}$, of $\mathcal{W}$ w.r.t. $\|\cdot\|_{2, X_{\text{pub}}}$, we can remove elements which do not lie in $\text{span}(S_{\text{pub}})$ and still have an $\alpha$-cover. Hence, the superset used in Algorithm 1, which essentially is, $\bar{\mathcal{W}} = P_{X_{\text{pub}}}\mathcal{W}$, is indeed an $\alpha$-cover.

The $n_{\text{pub}}$ we get is,

$$
\begin{aligned}
n_{\text{pub}} &= O\left( \max\left( \frac{R^2 \log(1/\beta)}{\alpha^2}, \min\left\{ m : \log^3(m)\mathfrak{R}_m^2(\mathcal{H}) \leq \alpha^2 \right\} \right) \right) \\
&= \tilde{O}\left( D^2 \|\mathcal{X}\|^2 \max\left( \frac{\log(2/\beta)}{\alpha^2}, \frac{1}{\alpha^2} \right) \right)
\end{aligned}
$$

where in the above, we plug in the Rademacher complexity of bounded linear predictor, $\mathfrak{R}_m(\mathcal{H}) = \Theta\left( \frac{D\|\mathcal{X}\|}{m} \right)$. Plugging the above in Equation (10),

$$
\begin{aligned}
&L(\widehat{w}; \mathcal{D}) - L(w^*; \mathcal{D}) \\
&= O\left( \frac{GD\|\mathcal{X}\|}{\sqrt{n_{\text{priv}}}} + \frac{GD\|\mathcal{X}\|\sqrt{\log(4/\beta)}}{\sqrt{n_{\text{priv}}}} + \frac{B\sqrt{\log(4/\beta)}}{\sqrt{n_{\text{priv}}}} \right) \\
&\quad + O\left( GD\|\mathcal{X}\| \left( \frac{\sqrt{n_{\text{pub}} \log(1/\delta)} + \sqrt{\log(4/\beta)}}{n_{\text{priv}}\epsilon} \right) \right) + G\alpha \\
&= O\left( \frac{GD\|\mathcal{X}\|\sqrt{\log(4/\beta)}}{\sqrt{n_{\text{priv}}}} + GD^2\|\mathcal{X}\|^2 \left( \frac{\sqrt{\log(2/\beta) + \log(1/\delta)}}{\alpha n_{\text{priv}}\epsilon} \right) + \frac{B\sqrt{\log(4/\beta)}}{\sqrt{n_{\text{priv}}}} \right) + G\alpha \\
&= O\left( GD\|\mathcal{X}\| \left( \frac{\sqrt{\log(4/\beta)}}{\sqrt{n_{\text{priv}}}} + \left( \frac{(\log(2/\beta) + \log(1/\delta))^{1/4}}{\sqrt{n_{\text{priv}}\epsilon}} \right) \right) + \frac{B\sqrt{\log(4/\beta)}}{\sqrt{n_{\text{priv}}}} \right)
\end{aligned}
$$

where the above follows by setting $\alpha = \frac{D\|\mathcal{X}\|(\log(1/\delta)+\log(2/\beta))^{1/4}}{\sqrt{n_{\text{priv}}\epsilon}}$. This yields the claimed rate. The resulting public sample complexity is

$$n_{\text{pub}} = \tilde{O}\left(D^2\|\mathcal{X}\|^2 \max\left(\frac{\log(2/\beta)}{\alpha^2}, \frac{1}{\alpha^2}\right)\right)$$

$$= \tilde{O}\left(\frac{n_{\text{priv}}\epsilon}{(\log(2/\beta)+\log(1/\delta))^{1/2}}\right).$$

This completes the proof.

**Lemma 6.** *Let $\tilde{\mathcal{H}}$ be a $\alpha$-cover of $\mathcal{H}$ with respect $\|\cdot\|_{2,X_{\text{pub}}}$. Then, $\bar{\mathcal{H}} = \tilde{\mathcal{H}} \cap span(X_{\text{pub}})$ is also an $\alpha$-cover.*

*Proof.* Given two $h_1, h_2 \in \mathcal{H}$, we have,

$$\|h_1 - h_2\|_{2,X_{\text{pub}}} = \sqrt{\frac{1}{n_{\text{pub}}} \sum_{i=1}^{n_{\text{pub}}} (h_1(x_i) - h_2(x_i))^2} = \frac{1}{\sqrt{n_{\text{pub}}}} \sqrt{(w_1 - w_2)^\top X_{\text{pub}}^\top X_{\text{pub}}(w_1 - w_2)}$$

where $w_1$ and $w_2$ are the vectors corresponding to linear functions $h_1$ and $h_2$ and $X_{\text{pub}}$ denote the matrix of public feature vectors.

Given any $h \in \mathcal{H}$, let $\tilde{h}$ denote the element closest to it in the cover $\tilde{\mathcal{H}}$; we have, $\|h - \tilde{h}\|_{2,X_{\text{pub}}} \leq \alpha$. Consider the singular value decomposition, $X_{\text{pub}} = V\Sigma U^\top$, where $U$ and $V$ are orthogonal matrices and $\Sigma$ is a diagonal matrix. Define $\bar{h} = P_{X_{\text{pub}}}(\tilde{h}) = UU^\top \tilde{h}$. Note that $U$ is an orthogonal projection onto $span(X_{\text{pub}})$ and $P_{X_{\text{pub}}}$ is the corresponding projection matrix. We have,

$$\|h - \bar{h}\|_{2,X_{\text{pub}}}^2 = \frac{1}{n_{\text{pub}}}(h - \bar{h})^\top X_{\text{pub}}^\top X_{\text{pub}}(h - \bar{h})$$

$$= \frac{1}{n_{\text{pub}}}(h - P_{X_{\text{pub}}}(\tilde{h}))^\top U\Sigma^2 U^\top (h - P_{X_{\text{pub}}}(\tilde{h}))$$

$$= \frac{1}{n_{\text{pub}}}(h - \tilde{h})^\top U(U^\top U)\Sigma^2(U^\top U)U^\top (h - \tilde{h})$$

$$= \frac{1}{n_{\text{pub}}}(h - \tilde{h})^\top U\Sigma^2 U^\top (h - \tilde{h})$$

$$= \frac{1}{n_{\text{pub}}}(h - \tilde{h})^\top X_{\text{pub}}^\top X_{\text{pub}}(h - \tilde{h})$$

$$\leq \alpha^2$$

Since by construction $\bar{h}$ also lies in $span(X_{\text{pub}})$, this proves the claim. $\qquad\square$

## C.2 Proof of Theorem 6

We state the complete version of this theorem and then present its proof.

**Theorem 14.** *Let $\epsilon > 0, \delta > 0$ and $\epsilon \leq \log(1/\delta)$. For a G-Lipschitz, B-bounded non-negative H-smooth loss function, Algorithm 1 satisfies $(\epsilon, \delta)$-DP. If the private sub-routine $\tilde{A}$ guarantees Equation (1) with probability at least $1 - \beta$, then with $n_{pub} = \tilde{O}\left(\frac{(HD\|\mathcal{X}\|)^{2/3}(n_{priv}\epsilon)^{2/3}}{G^{2/3}(\log(1/\delta))^{1/3}} + \frac{\sqrt{H}n_{priv}\epsilon\sqrt{L(w^*;\mathcal{D})}}{G\sqrt{\log(1/\delta)}}\right)$, with probability at least $1 - \beta$, $L(\hat{w};\mathcal{D}) - L(w^*;\mathcal{D})$ is at most*

$$\tilde{O}\left(\left(\frac{\sqrt{H}D\|\mathcal{X}\|}{\sqrt{n_{priv}\epsilon}} + \sqrt{\frac{B\log(8/\beta)}{n_{priv}}}\right)\sqrt{L(w^*;\mathcal{D})} + \frac{H\|\mathcal{X}\|^2 D^2}{n_{priv}\epsilon} + \frac{B\log(8/\beta)}{n_{priv}} + \frac{GD\|\mathcal{X}\|\sqrt{\log(4/\beta)}}{n_{priv}\epsilon}\right)$$

$$+ \tilde{O}\left(\left(\frac{\sqrt{H}D^2\|\mathcal{X}\|^2 G\sqrt{\log(1/\delta)}}{n_{priv}\epsilon}\right)^{2/3} + \frac{H^{1/4}D\|\mathcal{X}\|\sqrt{G}(\log(1/\delta))^{1/4}L(w^*;\mathcal{D})^{1/4}}{\sqrt{n_{priv}\epsilon}}\right)$$

*Further, with* $n_{pub} = \tilde{O}\left(\frac{(HD\|\mathcal{X}\|)^{2/3}(n_{priv}\epsilon)^{2/3}}{G^{2/3}(\log(1/\delta))^{1/3}} + \frac{\sqrt{H}n_{priv}\epsilon\sqrt{\widehat{L}(\widehat{w}^*;S_{priv})}}{G\sqrt{\log(1/\delta)}}\right)$, *with probability at least*

$1 - \beta$, *for any* $\bar{w} \in \mathcal{W}$, $L(\widehat{w};\mathcal{D}) - \widehat{L}(\widehat{w}^*;S_{priv})$ *is at most*

$$\tilde{O}\left(\left(\frac{\sqrt{H}D\|\mathcal{X}\|}{\sqrt{n_{priv}}\epsilon} + \sqrt{\frac{B\log(8/\beta)}{n_{priv}}}\right)\sqrt{\widehat{L}(\widehat{w}^*;S_{priv})} + \frac{H\|\mathcal{X}\|^2 D^2}{n_{priv}\epsilon} + \frac{B\log(8/\beta)}{n_{priv}} + \frac{GD\|\mathcal{X}\|\sqrt{\log(4/\beta)}}{n_{priv}\epsilon}\right)$$

$$+ \tilde{O}\left(\left(\frac{\sqrt{H}D^2\|\mathcal{X}\|^2 G\sqrt{\log(1/\delta)}}{n_{priv}\epsilon}\right)^{2/3} + \frac{H^{1/4}D\|\mathcal{X}\|\sqrt{G}(\log(1/\delta))^{1/4}\widehat{L}(\bar{w};S_{priv})^{1/4}}{\sqrt{n_{priv}}\epsilon}\right).$$

*where* $w^*$ *and* $\widehat{w}^*$ *are population and empirical minimizers with respect to* $\mathcal{D}$ *and* $S_{priv}$ *respectively.*

*Proof of Theorem 14.* The privacy guarantee follows from the privacy guarantee of sub-routine $\tilde{\mathcal{A}}$. The proof of the utility guarantee proceeds similar to that of Theorem 3. We define $w^* \in \arg\min_{w\in\mathcal{W}} L(w;\mathcal{D})$ and $\tilde{w}^* \in \arg\min_{w\in\tilde{\mathcal{W}}} L(w;U^\top\mathcal{D})$. Let $\mathring{w}^* \in \mathcal{W}$ such that $\mathring{w}^* = U\tilde{w}^*$. From the GLM structure, $L(\tilde{w}^*;U^\top\mathcal{D}) = L(\mathring{w}^*;\mathcal{D})$. We have,

$$\begin{aligned}
L(\widehat{w};\mathcal{D}) - L(w^*;\mathcal{D}) &= L(\widehat{w};\mathcal{D}) - \widehat{L}(\widehat{w};S_{priv}) + \widehat{L}(\widehat{w};S_{priv}) - L(w^*;\mathcal{D}) \\
&\leq L(\widehat{w};\mathcal{D}) - \widehat{L}(\widehat{w};S_{priv}) + \widehat{L}(\mathring{w}^*;S_{priv}) - L((\mathring{w}^*;\mathcal{D}) \\
&\quad + L(\mathring{w}^*;\mathcal{D}) - L(w^*;\mathcal{D}) + \widehat{L}(\widehat{w};S_{priv}) - \widehat{L}(\mathring{w}^*;S_{priv}) \\
&\leq \left|L(\widehat{w};\mathcal{D}) - \widehat{L}(\widehat{w};S_{priv})\right| + \left|L(\mathring{w}^*;\mathcal{D}) - \widehat{L}(\mathring{w}^*;S_{priv})\right| \\
&\quad + L(\mathring{w}^*;\mathcal{D}) - L(w^*;\mathcal{D}) + \widehat{L}(\tilde{w};\tilde{S}_{priv}) - \min_{w\in\tilde{\mathcal{W}}}\widehat{L}(w;\tilde{S}_{priv}) \quad (11)
\end{aligned}$$

The last term $\widehat{L}(\tilde{w};\tilde{S}_{priv}) - \min_{w\in\tilde{\mathcal{W}}}\widehat{L}(w;\tilde{S}_{priv})$ is bounded by the guarantees of the private sub-routine with probability at least $1 - \beta/4$,

$$\widehat{L}(\widehat{w};\tilde{S}_{priv}) - \min_{w\in\tilde{\mathcal{W}}}\widehat{L}(w;\tilde{S}_{priv}) = \tilde{O}\left(GD\|\mathcal{X}\|\left(\frac{\sqrt{n_{pub}\log(1/\delta)} + \sqrt{\log(4/\beta)}}{n_{priv}\epsilon}\right)\right). \quad (12)$$

To bound the term $L(\mathring{w}^*;\mathcal{D}) - L(w^*;\mathcal{D})$ in Equation (11), we apply smoothness to get,

$$\begin{aligned}
&L(\mathring{w}^*;\mathcal{D}) - L(w^*;\mathcal{D}) \\
&\leq L(\bar{w}^*;\mathcal{D}) - L(w^*;\mathcal{D}) \\
&\leq \mathbb{E}\left[\langle\phi_y'(\langle w^*,x\rangle),\langle\bar{w}^*,x\rangle - \langle w^*,x\rangle\rangle + \frac{H}{2}|\langle\bar{w}^*,x\rangle - \langle w^*,x\rangle|^2\right] \\
&\leq \mathbb{E}\left[|\phi_y'(\langle w^*,x\rangle)||\langle\bar{w}^*,x\rangle - \langle w^*,x\rangle| + \frac{H}{2}|\langle\bar{w}^*,x\rangle - \langle w^*,x\rangle|^2\right] \\
&\leq \sqrt{\mathbb{E}|\phi_y'(\langle w^*,x\rangle)|^2}\sqrt{\mathbb{E}_{x\sim\mathcal{D}_\mathcal{X}}|\langle\bar{w}^*,x\rangle - \langle w^*,x\rangle|^2} + \frac{H}{2}\mathbb{E}_{x\sim\mathcal{D}_\mathcal{X}}|\langle\bar{w}^*,x\rangle - \langle w^*,x\rangle|^2 \\
&\leq 2\sqrt{H\mathbb{E}_{x\sim\mathcal{D}}\phi_y(\langle w^*,x\rangle)}\sqrt{\mathbb{E}|\langle\bar{w}^*,x\rangle - \langle w^*,x\rangle|^2} + \frac{H}{2}\mathbb{E}_{x\sim\mathcal{D}_\mathcal{X}}|\langle\bar{w}^*,x\rangle - \langle w^*,x\rangle|^2 \\
&\leq 2\sqrt{HL(w^*;\mathcal{D})}\alpha + H\alpha^2 \quad (13)
\end{aligned}$$

where the above holds for any $\bar{w}^* \in \mathcal{H}$ such that $\bar{w}^* \in U\tilde{\mathcal{W}}$ by optimality of $\tilde{w}^*$ in $\tilde{\mathcal{W}}$. The second inequality holds from $H$-smoothness, the third and fourth from Cauchy-Schwarz, the fifth from self-bounding property of smooth non-negative losses (Lemma 4.1 in [SST10]). The final step holds with probability $1 - \beta/2$ from Lemma 1 with $n_{pub} = O\left(\max\left(\frac{R^2\log(2/\beta)}{\alpha^2}, \min\{m:\log^3(m)\mathfrak{R}_m^2(\mathcal{H}) \leq \alpha^2\}\right)\right)$ together with that since $\tilde{\mathcal{W}}$ is an $\alpha$-cover

of $\mathcal{H}$, together with Lemma 6 which shows that $\tilde{\mathcal{W}}$ is a valid $\alpha$-cover of $\mathcal{W}$. Therefore, there exists $\bar{h}^* \in \tilde{\mathcal{H}}$ with $\left\|\bar{h}^* - h^*\right\|_{2, S_{\text{pub}}} \leq \alpha$.

Further, applying AM-GM inequality, we get

$$L(\mathring{w}^*; \mathcal{D}) \leq 2L(w^*; \mathcal{D}) + 2H\alpha^2 \tag{14}$$

The first two terms in Equation (11) are bound via uniform convergence for smooth non-negative losses, (Theorem 1 in [SST10]) and Bernstein's inequality as follows; with probability at least $1 - \beta/4$, we have,

$$
\left| L(\widehat{w}; \mathcal{D}) - \widehat{L}(\widehat{w}; S_{\text{priv}}) \right| + \left| L(\mathring{w}^*; \mathcal{D}) - \widehat{L}(\mathring{w}^*; S_{\text{priv}}) \right|
$$

$$
= \tilde{O}\left( \sqrt{H}\mathfrak{R}_{n_{\text{priv}}}(\mathcal{H}) + \sqrt{\frac{B\log(8/\beta)}{n_{\text{priv}}}} \right) \left( \sqrt{\widehat{L}(\widehat{w}; S_{\text{priv}})} + \sqrt{L(\mathring{w}^*; \mathcal{D})} \right)
$$

$$
+ \tilde{O}\left( H\mathfrak{R}^2_{n_{\text{priv}}}(\mathcal{H}) + \frac{B\log(8/\beta)}{n_{\text{priv}}} \right)
$$

$$
= \tilde{O}\left( \frac{\sqrt{H}D\|\mathcal{X}\|}{\sqrt{n_{\text{priv}}}\epsilon} + \sqrt{\frac{B\log(8/\beta)}{n_{\text{priv}}}} \right) \left( \sqrt{\widehat{L}(\mathring{w}^*; S_{\text{priv}})} + \sqrt{L(\mathring{w}^*; \mathcal{D})} \right)
$$

$$
+ \tilde{O}\left( \frac{H\|\mathcal{X}\|^2 D^2}{n_{\text{priv}}\epsilon} + \frac{B\log(8/\beta)}{n_{\text{priv}}} \right) + \tilde{O}\left( GD\|\mathcal{X}\| \left( \frac{\sqrt{n_{\text{pub}}\log(1/\delta)} + \sqrt{\log(4/\beta)}}{n_{\text{priv}}\epsilon} \right) \right)
$$

$$
= \tilde{O}\left( \frac{\sqrt{H}D\|\mathcal{X}\|}{\sqrt{n_{\text{priv}}}\epsilon} + \sqrt{\frac{B\log(8/\beta)}{n_{\text{priv}}}} \right) \sqrt{L(\mathring{w}^*; \mathcal{D})}
$$

$$
+ \tilde{O}\left( \frac{H\|\mathcal{X}\|^2 D^2}{n_{\text{priv}}\epsilon} + \frac{B\log(8/\beta)}{n_{\text{priv}}} \right) + \tilde{O}\left( GD\|\mathcal{X}\| \left( \frac{\sqrt{n_{\text{pub}}\log(1/\delta)} + \sqrt{\log(4/\beta)}}{n_{\text{priv}}\epsilon} \right) \right)
$$

$$
= \tilde{O}\left( \frac{\sqrt{H}D\|\mathcal{X}\|}{\sqrt{n_{\text{priv}}}\epsilon} + \sqrt{H}\alpha + \sqrt{\frac{B\log(8/\beta)}{n_{\text{priv}}}} \right) \sqrt{L(w^*; \mathcal{D})}
$$

$$
+ \tilde{O}\left( \frac{H\|\mathcal{X}\|^2 D^2}{n_{\text{priv}}\epsilon} + H\alpha^2 + \frac{B\log(8/\beta)}{n_{\text{priv}}} \right) + O\left( GD\|\mathcal{X}\| \left( \frac{\sqrt{n_{\text{pub}}\log(1/\delta)} + \sqrt{\log(4/\beta)}}{n_{\text{priv}}\epsilon} \right) \right)
$$

$$\tag{15}$$

where the second equality follows from Equation (12), instantiating the Rademacher complexity of linear predictors, concavity of $x \mapsto \sqrt{x}$ and AM-GM inequality. The third equality follows concavity of $x \mapsto \sqrt{x}$ and Bernstein's inequality, the fourth follows from Equation (14) and AM-GM inequality.

Plugging the above, Equation (13) and Equation (12) into Equation (11), we get that with $n_{\text{pub}} = O\left( \max\left( \frac{\|\mathcal{X}\|^2 D^2 \log(2/\beta)}{\alpha^2}, \min\left\{ m : \log^3(n_{\text{pub}})\mathfrak{R}^2_{n_{\text{pub}}}(\mathcal{H}) \leq \alpha^2 \right\} \right) \right)$, the following holds with probability at least $1 - \beta$,

$$L(\widehat{w}; \mathcal{D}) - L(w^*; \mathcal{D})$$

$$= \tilde{O}\left( \frac{\sqrt{H}D\,\|\mathcal{X}\|}{\sqrt{n_{\text{priv}}}\epsilon} + \sqrt{H}\alpha + \sqrt{\frac{B\log{(8/\beta)}}{n_{\text{priv}}}} \right)\sqrt{L(w^*; \mathcal{D})}$$

$$+ \tilde{O}\left( \frac{H\,\|\mathcal{X}\|^2 D^2}{n_{\text{priv}}\epsilon} + H\alpha^2 + \frac{B\log{(8/\beta)}}{n_{\text{priv}}} \right) + O\left( GD\,\|\mathcal{X}\| \left( \frac{\sqrt{n_{\text{pub}}\log{(1/\delta)}} + \sqrt{\log{(4/\beta)}}}{n_{\text{priv}}\epsilon} \right) \right)$$

$$= \tilde{O}\left( \frac{\sqrt{H}D\,\|\mathcal{X}\|}{\sqrt{n_{\text{priv}}}\epsilon} + \frac{\sqrt{H}D\,\|\mathcal{X}\|}{\sqrt{n_{\text{pub}}}} + \sqrt{\frac{B\log{(8/\beta)}}{n_{\text{priv}}}} \right)\sqrt{L(w^*; \mathcal{D})}$$

$$+ \tilde{O}\left( \frac{H\,\|\mathcal{X}\|^2 D^2}{n_{\text{priv}}\epsilon} + \frac{HD^2\,\|\mathcal{X}\|^2}{n_{\text{pub}}} + \frac{B\log{(8/\beta)}}{n_{\text{priv}}} \right) + O\left( GD\,\|\mathcal{X}\| \left( \frac{\sqrt{n_{\text{pub}}\log{(1/\delta)}} + \sqrt{\log{(4/\beta)}}}{n_{\text{priv}}\epsilon} \right) \right)$$

$$\tag{16}$$

$$= \tilde{O}\left( \frac{\sqrt{H}D\,\|\mathcal{X}\|}{\sqrt{n_{\text{priv}}}\epsilon} + \sqrt{\frac{B\log{(8/\beta)}}{n_{\text{priv}}}} \right)\sqrt{L(w^*; \mathcal{D})}$$

$$+ \tilde{O}\left( \frac{H\,\|\mathcal{X}\|^2 D^2}{n_{\text{priv}}\epsilon} + \frac{B\log{(8/\beta)}}{n_{\text{priv}}} \right) + O\left( \frac{GD\,\|\mathcal{X}\|\sqrt{\log{(4/\beta)}}}{n_{\text{priv}}\epsilon} \right)$$

$$+ O\left( \left( \frac{\sqrt{H}D^2\,\|\mathcal{X}\|^2 G\sqrt{\log{(1/\delta)}}}{n_{\text{priv}}\epsilon} \right)^{2/3} + \frac{H^{1/4}D\,\|\mathcal{X}\|\sqrt{G}\left(\log{(1/\delta)}\right)^{1/4} L(w^*; \mathcal{D})^{1/4}}{\sqrt{n_{\text{priv}}\epsilon}} \right)$$

The public sample complexity is,

$$n_{\text{pub}} = \tilde{O}\left( \frac{(HD\,\|\mathcal{X}\|)^{2/3}(n_{\text{priv}}\epsilon)^{2/3}}{G^{2/3}(\log{(1/\delta)})^{1/3}} + \frac{\sqrt{H}n_{\text{priv}}\epsilon\sqrt{L(w^*; \mathcal{D})}}{G\sqrt{\log{(1/\delta)}}} \right)$$

This completes the first part of the theorem. For thee second part, we start from Equation (16),

$$L(\widehat{w};\mathcal{D}) \leq L(w^*;\mathcal{D}) + \tilde{O}\left(\frac{\sqrt{H}D\,\|\mathcal{X}\|}{\sqrt{n_{\mathrm{priv}}}\epsilon} + \frac{\sqrt{H}D\,\|\mathcal{X}\|}{\sqrt{n_{\mathrm{pub}}}} + \sqrt{\frac{B\log(8/\beta)}{n_{\mathrm{priv}}}}\right)\sqrt{L(w^*;\mathcal{D})}$$

$$+ \tilde{O}\left(\frac{H\,\|\mathcal{X}\|^2\,D^2}{n_{\mathrm{priv}}\epsilon} + \frac{HD^2\,\|\mathcal{X}\|^2}{n_{\mathrm{pub}}} + \frac{B\log(8/\beta)}{n_{\mathrm{priv}}}\right)$$

$$+ O\left(GD\,\|\mathcal{X}\|\left(\frac{\sqrt{n_{\mathrm{pub}}\log(1/\delta)} + \sqrt{\log(4/\beta)}}{n_{\mathrm{priv}}\epsilon}\right)\right)$$

$$\leq L(\widehat{w}^*;\mathcal{D}) + \tilde{O}\left(\frac{\sqrt{H}D\,\|\mathcal{X}\|}{\sqrt{n_{\mathrm{priv}}}\epsilon} + \frac{\sqrt{H}D\,\|\mathcal{X}\|}{\sqrt{n_{\mathrm{pub}}}} + \sqrt{\frac{B\log(8/\beta)}{n_{\mathrm{priv}}}}\right)\sqrt{L(\widehat{w}^*;\mathcal{D})}$$

$$+ \tilde{O}\left(\frac{H\,\|\mathcal{X}\|^2\,D^2}{n_{\mathrm{priv}}\epsilon} + \frac{HD^2\,\|\mathcal{X}\|^2}{n_{\mathrm{pub}}} + \frac{B\log(8/\beta)}{n_{\mathrm{priv}}}\right)$$

$$+ O\left(GD\,\|\mathcal{X}\|\left(\frac{\sqrt{n_{\mathrm{pub}}\log(1/\delta)} + \sqrt{\log(4/\beta)}}{n_{\mathrm{priv}}\epsilon}\right)\right)$$

$$\leq \widehat{L}(\widehat{w}^*;S_{\mathrm{priv}}) + \tilde{O}\left(\frac{\sqrt{H}D\,\|\mathcal{X}\|}{\sqrt{n_{\mathrm{priv}}}\epsilon} + \frac{\sqrt{H}D\,\|\mathcal{X}\|}{\sqrt{n_{\mathrm{pub}}}} + \sqrt{\frac{B\log(8/\beta)}{n_{\mathrm{priv}}}}\right)\sqrt{\widehat{L}(\widehat{w}^*;S_{\mathrm{priv}})}$$

$$+ \tilde{O}\left(\frac{H\,\|\mathcal{X}\|^2\,D^2}{n_{\mathrm{priv}}\epsilon} + \frac{HD^2\,\|\mathcal{X}\|^2}{n_{\mathrm{pub}}} + \frac{B\log(8/\beta)}{n_{\mathrm{priv}}}\right)$$

$$+ O\left(GD\,\|\mathcal{X}\|\left(\frac{\sqrt{n_{\mathrm{pub}}\log(1/\delta)} + \sqrt{\log(4/\beta)}}{n_{\mathrm{priv}}\epsilon}\right)\right)$$

$$\leq \widehat{L}(\widehat{w}^*;S_{\mathrm{priv}}) + \tilde{O}\left(\frac{\sqrt{H}D\,\|\mathcal{X}\|}{\sqrt{n_{\mathrm{priv}}}\epsilon} + \sqrt{\frac{B\log(8/\beta)}{n_{\mathrm{priv}}}}\right)\sqrt{\widehat{L}(\widehat{w}^*;\mathcal{D})}$$

$$+ \tilde{O}\left(\frac{H\,\|\mathcal{X}\|^2\,D^2}{n_{\mathrm{priv}}\epsilon} + \frac{B\log(8/\beta)}{n_{\mathrm{priv}}}\right) + O\left(\frac{GD\,\|\mathcal{X}\|\,\sqrt{\log(4/\beta)}}{n_{\mathrm{priv}}\epsilon}\right)$$

$$+ O\left(\left(\frac{\sqrt{H}D^2\,\|\mathcal{X}\|^2\,G\sqrt{\log(1/\delta)}}{n_{\mathrm{priv}}\epsilon}\right)^{2/3} + \frac{H^{1/4}D\,\|\mathcal{X}\|\,\sqrt{G}\,(\log(1/\delta))^{1/4}\,L(\widehat{w};\mathcal{D})^{1/4}}{\sqrt{n_{\mathrm{priv}}}\epsilon}\right)$$

where the second inequality holds form optimality of $w*$, the third from uniform convergence, Theorem 1 in [SST10] and AM-GM inequality, and the last by plugging in the following public sample complexity.

$$n_{\mathrm{pub}} = \tilde{O}\left(\frac{(HD\,\|\mathcal{X}\|)^{2/3}(n_{\mathrm{priv}}\epsilon)^{2/3}}{G^{2/3}(\log(1/\delta))^{1/3}} + \frac{\sqrt{H}n_{\mathrm{priv}}\epsilon\sqrt{\widehat{L}(\widehat{w}^*;S_{\mathrm{priv}})}}{G\sqrt{\log(1/\delta)}}\right)$$

This completes the proof.

$\square$

**Theorem 15.** *In the setting of Theorem 14 with the additional assumption that the global minimizer of risk $L$, $w^*$ lies in $\mathcal{W}$, we get that with $n_{pub} = \tilde{O}\left(\frac{(HD\|\mathcal{X}\|)^{2/3}(n_{priv}\epsilon)^{2/3}}{G^{2/3}(\log(1/\delta))^{1/3}}\right)$, with probability at least*

$1 - \beta,$

$$L(\widehat{w}; \mathcal{D}) - L(w^*; \mathcal{D})$$

$$= \tilde{O}\left(\frac{\sqrt{H}D\,\|\mathcal{X}\|}{\sqrt{n_{priv}}\epsilon} + \sqrt{\frac{B\log(8/\beta)}{n_{priv}}}\right)\sqrt{L(w^*; \mathcal{D})} + \tilde{O}\left(\frac{H\,\|\mathcal{X}\|^2\,D^2}{n_{priv}\epsilon} + \frac{B\log(8/\beta)}{n_{priv}}\right)$$

$$+ O\left(\frac{GD\,\|\mathcal{X}\|\,\sqrt{\log(4/\beta)}}{n_{priv}\epsilon}\right) + O\left(\left(\frac{\sqrt{H}D^2\,\|\mathcal{X}\|^2\,G\sqrt{\log(1/\delta)}}{n_{priv}\epsilon}\right)^{2/3}\right)$$

*Further,*

$$L(\widehat{w}; \mathcal{D}) - \widehat{L}(\widehat{w}^*; S_{priv})$$

$$= \tilde{O}\left(\frac{\sqrt{H}D\,\|\mathcal{X}\|}{\sqrt{n_{priv}}\epsilon} + \sqrt{\frac{B\log(8/\beta)}{n_{priv}}}\right)\sqrt{\widehat{L}(\widehat{w}^*; S_{priv})} + \tilde{O}\left(\frac{H\,\|\mathcal{X}\|^2\,D^2}{n_{priv}\epsilon} + \frac{B\log(8/\beta)}{n_{priv}}\right)$$

$$+ O\left(\frac{GD\,\|\mathcal{X}\|\,\sqrt{\log(4/\beta)}}{n_{priv}\epsilon}\right) + O\left(\left(\frac{\sqrt{H}D^2\,\|\mathcal{X}\|^2\,G\sqrt{\log(1/\delta)}}{n_{priv}\epsilon}\right)^{2/3}\right).$$

*where $w^*$ and $\widehat{w}^*$ are population and empirical minimizers with respect to $\mathcal{D}$ and $S_{priv}$ respectively.*

*Proof.* The proof is almost identical to that of Theorem 15. We repeat the steps pointing out the differences and how the expressions change. We continue till Equation (12). Next, we apply smoothness which results in the key difference between the analyses,

$$L(\mathring{w}^*; \mathcal{D}) - L(w^*; \mathcal{D})$$
$$\leq L(\bar{w}^*; \mathcal{D}) - L(w^*; \mathcal{D})$$
$$\leq \mathbb{E}\left[\langle \phi'_y(\langle w^*, x\rangle), \langle \bar{w}^*, x\rangle - \langle w^*, x\rangle\rangle + \frac{H}{2}|\langle \bar{w}^*, x\rangle - \langle w^*, x\rangle|^2\right]$$
$$\leq \langle \mathbb{E}\left[\phi'_y(\langle w^*, x\rangle)x\right], \bar{w}^* - w^*\rangle + \frac{H}{2}\mathbb{E}\left[|\langle \bar{w}^*, x\rangle - \langle w^*, x\rangle|^2\right]$$
$$\leq \langle \nabla L(w^*; \mathcal{D}), \bar{w}^* - w^*\rangle + H\alpha^2$$
$$= H\alpha^2$$

where last equality uses the fact that $\nabla L(w^*; \mathcal{D}) = 0$ since $w^*$ is the unconstrained minimizer. Continuing, we get,

$$\left|L(\widehat{w}; \mathcal{D}) - \widehat{L}(\widehat{w}; S_{\text{priv}})\right| + \left|L(\mathring{w}^*; \mathcal{D}) - \widehat{L}(\mathring{w}^*; S_{\text{priv}})\right|$$

$$= \tilde{O}\left(\frac{\sqrt{H}D\,\|\mathcal{X}\|}{\sqrt{n_{\text{priv}}}\epsilon} + \sqrt{\frac{B\log(8/\beta)}{n_{\text{priv}}}}\right)\sqrt{L(w^*; \mathcal{D})}$$

$$+ \tilde{O}\left(\frac{H\,\|\mathcal{X}\|^2\,D^2}{n_{\text{priv}}\epsilon} + \frac{B\log(8/\beta)}{n_{\text{priv}}}\right) + O\left(GD\,\|\mathcal{X}\|\left(\frac{\sqrt{n_{\text{pub}}\log(1/\delta)} + H\alpha^2 + \sqrt{\log(4/\beta)}}{n_{\text{priv}}\epsilon}\right)\right)$$

This yields,

$L(\widehat{w}; \mathcal{D}) - L(w^*; \mathcal{D})$

$$= \tilde{O}\left( \frac{\sqrt{H}D\,\|\mathcal{X}\|}{\sqrt{n_{\text{priv}}}\epsilon} + \sqrt{\frac{B\log(8/\beta)}{n_{\text{priv}}}} \right)\sqrt{L(w^*; \mathcal{D})}$$

$$+ \tilde{O}\left( \frac{H\,\|\mathcal{X}\|^2 D^2}{n_{\text{priv}}\epsilon} + H\alpha^2 + \frac{B\log(8/\beta)}{n_{\text{priv}}} \right) + O\left( GD\,\|\mathcal{X}\| \left( \frac{\sqrt{n_{\text{pub}}\log(1/\delta)} + \sqrt{\log(4/\beta)}}{n_{\text{priv}}\epsilon} \right) \right)$$

$$= \tilde{O}\left( \frac{\sqrt{H}D\,\|\mathcal{X}\|}{\sqrt{n_{\text{priv}}}\epsilon} + \sqrt{\frac{B\log(8/\beta)}{n_{\text{priv}}}} \right)\sqrt{L(w^*; \mathcal{D})}$$

$$+ \tilde{O}\left( \frac{H\,\|\mathcal{X}\|^2 D^2}{n_{\text{priv}}\epsilon} + \frac{HD^2\,\|\mathcal{X}\|^2}{n_{\text{pub}}} + \frac{B\log(8/\beta)}{n_{\text{priv}}} \right) + O\left( GD\,\|\mathcal{X}\| \left( \frac{\sqrt{n_{\text{pub}}\log(1/\delta)} + \sqrt{\log(4/\beta)}}{n_{\text{priv}}\epsilon} \right) \right)$$

$$= \tilde{O}\left( \frac{\sqrt{H}D\,\|\mathcal{X}\|}{\sqrt{n_{\text{priv}}}\epsilon} + \sqrt{\frac{B\log(8/\beta)}{n_{\text{priv}}}} \right)\sqrt{L(w^*; \mathcal{D})} + \tilde{O}\left( \frac{H\,\|\mathcal{X}\|^2 D^2}{n_{\text{priv}}\epsilon} + \frac{B\log(8/\beta)}{n_{\text{priv}}} \right)$$

$$+ O\left( \frac{GD\,\|\mathcal{X}\|\sqrt{\log(4/\beta)}}{n_{\text{priv}}\epsilon} \right) + O\left( \left( \frac{\sqrt{H}D^2\,\|\mathcal{X}\|^2 G\sqrt{\log(1/\delta)}}{n_{\text{priv}}\epsilon} \right)^{2/3} \right)$$

The public sample complexity is,

$$n_{\text{pub}} = \tilde{O}\left( \frac{(HD\,\|\mathcal{X}\|)^{2/3}(n_{\text{priv}}\epsilon)^{2/3}}{G^{2/3}(\log(1/\delta))^{1/3}} \right).$$

The second part follows similarly.

$\square$

## C.3   Lower bounds

### C.3.1   Proof of Theorem 4

To establish the $\frac{GD\|\mathcal{X}\|}{\sqrt{n}}$ term in the lower bound, we consider a one-dimenisonal problem where the loss $\phi_y(\widehat{y}) = -Gy\widehat{y}$ and marginal distribution $\mathcal{D}_x$ as the point distribution on $\|\mathcal{X}\|$ such that the overall loss is $\underset{x,y}{\mathbb{E}}\left[\ell(w, (x, y))\right] = \underset{y}{\mathbb{E}}\left[y \cdot w\|\mathcal{X}\|G\right]$. We further set $\mathcal{W} = [-D, D]$ and consider $\mathcal{D}_y$ to be the distribution which as 1 with probability $\mathbb{P}\left[y = 1\right] = (1 + \mu)/2$ and $\mathbb{P}\left[y = 1\right] = (1 - \mu)/2$ for some $\mu \in [-1, 1]$. Note the minimizer $w^* = D\frac{\mu}{|\mu|}$ achieves population risk $-\mu GD\|\mathcal{X}\|$. Classic results in information theory establish if $\mu$ is sampled uniformly from $\{\pm\frac{1}{\sqrt{6n}}\}$, no algorithm can estimate the sign of $\mu$ with probability better than $1/2$ (see [Duc23, Section 8.3]). Thus it must be that for any algorithm $\mathbb{E}_{\mathcal{A},S}\left[L(\mathcal{A}(S); \mathcal{D}) - \min_{w \in \mathbb{R}^d} L(w; \mathcal{D})\right] = \Omega\left(\frac{GD\|\mathcal{X}\|}{\sqrt{n}}\right)$.

The $GD\,\|\mathcal{X}\|\min\left\{\frac{1}{\sqrt{n}\epsilon}, \frac{\sqrt{d}}{n\epsilon}\right\}$ term in the lower bound is essentially a corollary of [ABG$^+$22, Theorem 6]. We provide further remarks here. The loss function used is,

$$\ell(w; (x, y)) = \phi_y(\langle w, x \rangle) = |y - \langle w, x \rangle|.$$

Define $d' := \min(d, n\epsilon)$ and $p := \min\left(1, \frac{d'}{n\epsilon}\right)$. The (known) marginal distribution $\mathcal{D}_{\mathcal{X}}$ is described as: with probability $1 - p$, $x = \vec{0}$, otherwise, $x \sim \text{Unif}\left(\|\mathcal{X}\|\{e_j\}_{j=1}^{d'}\right)$ where $e_j$'s are canonical basis vectors. The (unknown) conditional distribution of the response $y$ is as follows. Sample a "fingerprint-ing code", $z' \in \{0, 1\}^{d'}$ with mean $\mu' \in [0, 1]^{d'}$ where each co-ordinate $\mu'_j \sim \text{Beta}(0.0625, 0.0625)$ i.i.d. Embed $z'$ in $d$ dimensions as $z$ and let $\mu$ be the corresponding mean vector. Finally, define $y = \frac{D\langle z, x \rangle}{\sqrt{d'}}$. The proof in [ABG$^+$22, Theorem 6] then proceeds by lower bounding the loss by bounding the ability of any differential private algorithm to estimate the fingerprinting code $z$.

Since the rank of $\mathbb{E}_{x \sim \mathcal{D}_{\mathcal{X}}}\left[xx^{\top}\right] = d'$, the result [ABG$^+$22, Theorem 6] then yields a lower bound on the *unconstrained excess risk*,

$$\mathbb{E}_{\mathcal{A},S}\left[L(\mathcal{A}(S);\mathcal{D}) - \min_{w \in \mathbb{R}^d} L(w;\mathcal{D})\right] = \Omega\left(GD\|\mathcal{X}\|\min\left\{\frac{1}{\sqrt{n\epsilon}}, \frac{\sqrt{d}}{n\epsilon}\right\}\right),$$

but also guarantees that the global minimizer has norm at most $D$. Thus, we achieve the same lower bound for $\mathbb{E}_{\mathcal{A},S}\left[L(\mathcal{A}(S);\mathcal{D}) - \min_{w \in \mathcal{W}} L(w;\mathcal{D})\right]$ by setting $\mathcal{W}$ to be the ball of radius $D$.

### C.3.2 Proof of Theorem 5

The proof uses the lower bound instance in the DP-SCO lower bound with public data, Theorem 1. We consider the case where $\mathcal{D}_y$ is the point distribution on 1. Then for any $y \in \mathcal{Y}$, $\mathcal{Y} = \{1\}$, the loss function is then $\ell(w;(x,y)) = y\langle w,x\rangle = \langle w,x\rangle$, as in Theorem 1. Hence, a labeled and unlabeled sample have the same information. We also set $\mathcal{W}$ to be the ball of radius $D$.

Assume by contradiction there exists an $(\epsilon,\delta)$-PA-DP algorithm, $\mathcal{A}$, which achieves rate $O\left(GD\|\mathcal{X}\|(\frac{1}{\sqrt{n_{\mathrm{priv}}}} + \frac{\sqrt{\log(1/\delta)}}{\sqrt{n_{\mathrm{priv}}\epsilon}})\right)$ with $o(n_{\mathrm{priv}}\epsilon/\log(1/\delta))$ public samples. Since $n_{\mathrm{pub}} = o(n_{\mathrm{priv}}\epsilon/\log(1/\delta))$ and $d = \omega(n\epsilon)$, Theorem 1 gives a lower bound on $\mathbb{E}\left[\mathcal{A}(X_{\mathrm{pub}}, S_{\mathrm{priv}};\mathcal{D}) - \min_{w \in \mathcal{W}}\{L(w;\mathcal{D})\}\right]$ of

$$\Omega\left(GD\|\mathcal{X}\|\min\left\{\frac{1}{\sqrt{n_{\mathrm{pub}}}}, \frac{\sqrt{d\log(1/\delta)}}{n_{\mathrm{priv}}\epsilon}\right\}\right) = \omega\left(GD\|\mathcal{X}\|\frac{\sqrt{\log(1/\delta)}}{\sqrt{n_{\mathrm{priv}}\epsilon}}\right).$$

Since $\epsilon \leq 1$, this is a contradiction.

## D Missing proofs for Section 4.2

### D.1 Proof of Theorem 7

*Proof.* The privacy proof follows from the guarantee of exponential mechanism [MT07]. In particular, the sensitivity of the score function is at most $\frac{2}{n_{\mathrm{priv}}}\min(B,GR)$ where the first follows from the loss bound of $B$ and the second from the Lipschitzness and bound on predictors. Let $h^* \in \arg\min_{h \in \mathcal{H}} L(h;\mathcal{D})$ and $\tilde{h}^* \in \arg\min_{h \in \tilde{\mathcal{H}}} L(h;\mathcal{D})$. From standard analysis based on uniform convergence, we have

$$
\begin{aligned}
L(\widehat{h};\mathcal{D}) - L(h^*;\mathcal{D}) &= L(\widehat{h};\mathcal{D}) - \widehat{L}(\widehat{h};S_{\mathrm{priv}}) + \widehat{L}(\widehat{h};S_{\mathrm{priv}}) - L(h^*;\mathcal{D})\\
&\leq \sup_{h \in \mathcal{H}}\left(L(h;\mathcal{D}) - \widehat{L}(h;S_{\mathrm{priv}})\right) + L(\tilde{h}^*;\mathcal{D}) - L(h^*;\mathcal{D})\\
&\quad + \widehat{L}(\tilde{h}^*;S_{\mathrm{priv}}) - L(\tilde{h}^*;\mathcal{D}) + \widehat{L}(\widehat{h};S_{\mathrm{priv}}) - \widehat{L}(\tilde{h}^*;S_{\mathrm{priv}}) \quad (17)\\
&\leq 2\sup_{h \in \mathcal{H}}\left|L(h;\mathcal{D}) - \widehat{L}(h;S_{\mathrm{priv}})\right| + L(\tilde{h}^*;\mathcal{D}) - L(h^*;\mathcal{D})\\
&\quad + \widehat{L}(\widehat{h};S_{\mathrm{priv}}) - \min_{h \in \tilde{\mathcal{H}}}\widehat{L}(h;S_{\mathrm{priv}}) \quad (18)\\
&\leq 2G\mathfrak{R}_{n_{\mathrm{priv}}}(\mathcal{H}) + O\left(\frac{B\sqrt{\log(4/\beta)}}{\sqrt{n_{\mathrm{priv}}}}\right)\\
&\quad + O\left(\frac{\min(B,GR)(\log(|\tilde{\mathcal{H}}|) + \log(4/\beta))}{n_{\mathrm{priv}}\epsilon}\right) + L(\tilde{h}^*;\mathcal{D}) - L(h^*;\mathcal{D})\\
&\qquad\qquad\qquad\qquad\qquad\qquad\qquad\qquad\qquad\qquad\qquad\qquad\qquad\qquad (19)
\end{aligned}
$$

where the above holds with probability at least $1 - \beta/2$ and follows from guarantee of exponential mechanism [MT07] and uniform convergence ([SSBD14], see Theorem 16). We further have that $\log\left(|\tilde{\mathcal{H}}|\right) = \tilde{O}(\mathrm{fat}_{c\alpha}(\mathcal{H}))$ from Lemma 7.

For the $L(\tilde{h}^*; \mathcal{D}) - L(h^*; \mathcal{D})$ term, we have

$$
\begin{aligned}
L(\tilde{h}^*; \mathcal{D}) - L(h^*; \mathcal{D}) &\leq L(\bar{h}^*; \mathcal{D}) - L(h^*; \mathcal{D}) \\
&\leq G\mathbb{E}\left|\bar{h}^*(x) - h^*(x)\right| \\
&\leq G\sqrt{\mathbb{E}\left|\bar{h}^*(x) - h^*(x)\right|^2} \\
&\leq 2G\alpha
\end{aligned}
$$

where the first step holds for any $\bar{h}^* \in \tilde{\mathcal{H}}$ by optimaility of $\tilde{h}^*$ over $\tilde{\mathcal{H}}$, the second holds from the $G$-Lipschitzness of the loss function, the third from Jensen's inequality. The final step holds with probability $1 - \beta/2$ from Lemma 1 with $n_{\text{pub}} = O\left(\max\left(\frac{R^2\log(2/\beta)}{\alpha^2}, \min\left\{m : \log^3(m)\mathfrak{R}_m^2(\mathcal{H}) \leq \alpha^2\right\}\right)\right)$ together with the fact that since $\tilde{\mathcal{H}}$ is an $\alpha$-cover of $\mathcal{H}$, hence there exists $\bar{h}^* \in \tilde{\mathcal{H}}$ with $\left\|\bar{h}^* - h^*\right\|_{2, X_{\text{pub}}} \leq \alpha$.

Plugging the above in Equation (19), we get with probability at least $1 - \beta$,

$$
\begin{aligned}
L(\widehat{h}; \mathcal{D}) - L(h^*; \mathcal{D}) &\leq 2G\mathfrak{R}_{n_{\text{priv}}}(\mathcal{H}) + O\left(\frac{B\sqrt{\log(4/\beta)}}{\sqrt{n_{\text{priv}}}}\right) \\
&\quad + O\left(\frac{\min(B, GR)(\text{fat}_{c\alpha}(\mathcal{H}) + \log(4/\beta))}{n_{\text{priv}}\epsilon}\right) + 2G\alpha,
\end{aligned}
$$

which finishes the proof. $\qquad\square$

**Theorem 16.** *[SSBD14] Let $\mathcal{H} \subseteq [-R, R]^{\mathcal{X}}$. For any $G$-Lipschitz, $B$-bounded loss function, any probability distribution $\mathcal{D}$ over $\mathcal{X} \times \mathcal{Y}$, given $m$ i.i.d. samples from $S$, with probability at least $1 - \beta$, the following holds for all $h \in \mathcal{H}$,*

$$
\left|L(h; \mathcal{D}) - \widehat{L}(h; S)\right| \leq 2G\mathfrak{R}_m(\mathcal{H}) + O\left(B\sqrt{\frac{\log(4/\beta)}{m}}\right)
$$

*Proof.* This is a classical result in learning theory which follows directly Theorem 26.5 in [SSBD14] together with the contraction lemma (Lemma 26.9 in [SSBD14]) for Lipschitz losses. $\qquad\square$

**Lemma 7.** *Let $\tilde{\mathcal{H}}$ be an $\alpha$-cover of $\mathcal{H}$ with respect to $\|\cdot\|_{2, X_{pub}}$. The size of $\tilde{\mathcal{H}}$ is bounded as,*

$$
\log\left(\left|\tilde{\mathcal{H}}\right|\right) \leq \text{fat}_{c\alpha}(\mathcal{H})\log\left(\frac{2R}{\alpha}\right)
$$

*where $c$ is an absolute constant.*

*Proof.* This follows directly from Theorem 8,

$$
\log\left(\left|\tilde{\mathcal{H}}\right|\right) = \mathcal{N}_2\left(\mathcal{H}, \alpha, S_{\text{pub}}\right) \leq \mathcal{N}_2\left(\mathcal{H}, \alpha, n_{\text{pub}}\right) \leq \text{fat}_{c\alpha}(\mathcal{H})\log\left(\frac{2R}{\alpha}\right).
$$

$\qquad\square$

**Proof of Lemma 1** For $h \in \mathcal{H}$, let $\tilde{h} \in \arg\min_{\bar{h} \in \tilde{\mathcal{H}}} \|\bar{h} - h\|_{2, X_{\text{pub}}}$. Since $\tilde{\mathcal{H}}$ is an $\tau$-cover, this gives us that $\|h - \tilde{h}\|_{2, X_{\text{pub}}} \leq \tau$. We have,

$$
\begin{aligned}
\|h - \tilde{h}\|_{2, \mathcal{D}_{\mathcal{X}}}^2 &= \mathbb{E}\left|h(x) - \tilde{h}(x)\right|^2 \\
&= \mathbb{E}\left|h(x) - \tilde{h}(x)\right|^2 - \frac{1}{n_{\text{pub}}}\sum_{x \in S_{\text{pub}}}\left|h(x) - \tilde{h}(x)\right|^2 + \frac{1}{n_{\text{pub}}}\sum_{x \in S_{\text{pub}}}\left|h(x) - \tilde{h}(x)\right|^2 \\
&\leq \sup_{\bar{h} \in \mathcal{H}}\left(\mathbb{E}\left|h(x) - \bar{h}(x)\right|^2 - \frac{1}{n_{\text{pub}}}\sum_{x \in S_{\text{pub}}}\left|h(x) - \bar{h}(x)\right|^2\right) + \tau^2.
\end{aligned}
$$

The first term above can be seen as uniform deviation between the empirical and population risk, of another prediction problem, with squared loss, in the the realizable setting (with the responses generated by $h$). The squared loss is $\frac{1}{2}$-smooth and non-negative, so we can apply result of Theorem 1 in [SST10] instantiated in the realizable setting, which gives us that with probability at least $1 - \beta$,

$$\|h - \tilde{h}\|^2_{2.\mathcal{D}_{\mathcal{X}}} = O\left(\log^3(n_{\text{pub}})\mathfrak{R}^2_{n_{\text{pub}}}(\mathcal{H}) + \frac{R^2 \log(1/\beta)}{n_{\text{pub}}}\right) + \tau^2.$$

Choosing $n_{\text{pub}}$ such that $n_{\text{pub}} = O\left(\max\left(\frac{R^2 \log(1/\beta)}{\alpha^2}, \min\left\{m : \log^3(m)\mathfrak{R}^2_m(\mathcal{H}) \leq \alpha^2\right\}\right)\right)$, we get the claimed result.

## D.2 Optimistic rates with smooth non-negative losses

Algorithm 2 achieves optimistic rates on risk depending on realizability/interpolation conditions, that is, whenever $L(h^*; \mathcal{D})$ or $\widehat{L}(\widehat{h}; S_{\text{priv}})$ is small.

**Theorem 17.** *Algorithm 2 with $\gamma = \frac{2B}{n_{priv}\epsilon}$ satisfies $\epsilon$-PA-DP. For $n_{pub} = O\left(\max\left(\frac{R^2 \log(2/\beta)}{\alpha^2}, \min\left\{m : log^3(m)\mathfrak{R}^2_m(\mathcal{H}) \leq \alpha^2\right\}\right)\right) < \infty$ and any $\alpha > 0$, with probability at least $1 - \beta$, we have*

$$L(\widehat{h}; \mathcal{D}) - L(h^*; \mathcal{D}) = \tilde{O}\left(\sqrt{H}\mathfrak{R}_{n_{priv}}(\mathcal{H}) + \sqrt{H}\alpha + \sqrt{\frac{B \log(8/\delta)}{n_{priv}}}\right)\sqrt{L(h^*; \mathcal{D})}$$

$$+ \tilde{O}\left(H\mathfrak{R}^2_{n_{priv}}(\mathcal{H}) + H\alpha^2 + \frac{B \log(8/\beta)}{n_{priv}} + \frac{B(fat_{c\alpha}(\mathcal{H}) + \log(4/\beta))}{n_{priv}\epsilon}\right)$$

$$L(\widehat{h}; \mathcal{D}) - \widehat{L}(\widehat{h}^*; S_{priv}) \leq \tilde{O}\left(\sqrt{H}\mathfrak{R}_{n_{priv}}(\mathcal{H}) + \sqrt{H}\alpha + \sqrt{\frac{B \log(8/\delta)}{n_{priv}}}\right)\sqrt{\widehat{L}(\widehat{h}^*; S_{priv})}$$

$$+ \tilde{O}\left(H\mathfrak{R}^2_{n_{priv}}(\mathcal{H}) + H\alpha^2 + \frac{B \log(8/\beta)}{n_{priv}} + \frac{B(fat_{c\alpha}(\mathcal{H}) + \log(4/\beta))}{n_{priv}\epsilon}\right)$$

*where $h^*$ and $\widehat{h}^*$ are population and empirical minimizers with respect to $\mathcal{D}$ and $S_{priv}$ respectively, and $c$ is an absolute constant.*

*Proof of Theorem 17.* The privacy proof is the same as that of Theorem 7. For utility, let $h^* \in \arg\min_{h\in\mathcal{H}} L(h; \mathcal{D})$ and $\tilde{h}^* \in \arg\min_{h\in\tilde{\mathcal{H}}} L(h; \mathcal{D})$.

We start with the proof of the first part of the theorem. We have,

$$L(\widehat{h}; \mathcal{D}) - L(h^*; \mathcal{D}) = L(\widehat{h}; \mathcal{D}) - \widehat{L}(\widehat{h}; S_{\text{priv}}) + \widehat{L}(\widehat{h}; S_{\text{priv}}) - L(h^*; \mathcal{D})$$

$$\leq L(\widehat{h}; \mathcal{D}) - \widehat{L}(\widehat{h}; S_{\text{priv}}) + \widehat{L}(\tilde{h}^*; S_{\text{priv}}) - L(\tilde{h}^*; \mathcal{D})$$

$$+ L(\tilde{h}^*; \mathcal{D}) - L(h^*; \mathcal{D}) + \widehat{L}(\widehat{h}; S_{\text{priv}}) - \widehat{L}(\tilde{h}^*; S_{\text{priv}})$$

$$\leq \left|L(\widehat{h}; \mathcal{D}) - \widehat{L}(\widehat{h}; S_{\text{priv}})\right| + \left|L(\tilde{h}^*; \mathcal{D}) - \widehat{L}(\tilde{h}^*; S_{\text{priv}})\right|$$

$$+ L(\tilde{h}^*; \mathcal{D}) - L(h^*; \mathcal{D}) + \widehat{L}(\widehat{h}; S_{\text{priv}}) - \widehat{L}(\tilde{h}^*; S_{\text{priv}}) \qquad (20)$$

From the guarantee of exponential mechanism together with $\log\left(\left|\tilde{\mathcal{H}}\right|\right) = \tilde{O}(fat_{c\alpha}(\mathcal{H}))$ from Lemma 7, we have that with probability at least $1 - \beta/4$,

$$\widehat{L}(\widehat{h}; S_{\text{priv}}) - \widehat{L}(\tilde{h}^*; S_{\text{priv}}) \leq \widehat{L}(\widehat{h}; S_{\text{priv}}) - \min_{h\in\tilde{\mathcal{H}}} \widehat{L}(h; S_{\text{priv}}) = O\left(\frac{B(\log\left(\left|\tilde{\mathcal{H}}\right|\right) + \log(4/\beta))}{n_{\text{priv}}\epsilon}\right)$$

$$= \tilde{O}\left(\frac{B(fat_{c\alpha}(\mathcal{H}) + \log(4/\beta))}{n_{\text{priv}}\epsilon}\right) \qquad (21)$$

For the $L(\tilde{h}^*; \mathcal{D}) - L(h^*; \mathcal{D})$ term in Equation (20), we apply smoothness to get,

$$L(\tilde{h}^*; \mathcal{D}) - L(h^*; \mathcal{D}) \leq L(\bar{h}^*; \mathcal{D}) - L(h^*; \mathcal{D})$$

$$\leq \mathbb{E}\left[\langle \phi_y'(h^*(x)), \bar{h}^*(x) - h^*(x)\rangle + \frac{H}{2}\left|\bar{h}^*(x) - h^*(x)\right|^2\right]$$

$$\leq \mathbb{E}\left[\left|\phi_y'(h^*(x))\right|\left|\bar{h}^*(x) - h^*(x)\right| + \frac{H}{2}\left|\bar{h}^*(x) - h^*(x)\right|^2\right]$$

$$\leq \sqrt{\mathbb{E}\left|\phi_y'(h^*(x))\right|^2}\sqrt{\mathbb{E}_{x \sim \mathcal{D}_{\mathcal{X}}}\left|\bar{h}^*(x) - h^*(x)\right|^2} + \frac{H}{2}\mathbb{E}_{x \sim \mathcal{D}_{\mathcal{X}}}\left|\bar{h}^*(x) - h^*(x)\right|^2$$

$$\leq 2\sqrt{H\mathbb{E}_{x \sim \mathcal{D}}\phi_y(h^*(x))}\sqrt{\mathbb{E}\left|\bar{h}^*(x) - h^*(x)\right|^2} + \frac{H}{2}\mathbb{E}_{x \sim \mathcal{D}_{\mathcal{X}}}\left|\bar{h}^*(x) - h^*(x)\right|^2$$

$$\leq 2\sqrt{HL(h^*; \mathcal{D})}\alpha + H\alpha^2 \tag{22}$$

where the above holds for any $\bar{h}^* \in \tilde{\mathcal{H}}$. The second inequality holds from $H$-smoothness, the third and fourth from Cauchy-Schwarz, the fifth from self-bounding property of smooth non-negative losses [SST10]. The final step holds with probability $1 - \beta/2$ from Lemma 1 with $n_{\text{pub}} = O\left(\max\left(\frac{R^2 \log(2/\beta)}{\alpha^2}, \min\left\{m : \log^3(m)\mathfrak{R}_m^2(\mathcal{H}) \leq \alpha^2\right\}\right)\right)$ together with the fact that since $\tilde{\mathcal{H}}$ is an $\alpha$-cover of $\mathcal{H}$, so there exists $\bar{h}^* \in \tilde{\mathcal{H}}$ with $\left\|\bar{h}^* - h^*\right\|_{2, X_{\text{pub}}} \leq \alpha$.

An application of AM-GM inequality further yields,

$$L(\tilde{h}^*; \mathcal{D}) \leq 2L(h^*; \mathcal{D}) + 2H\alpha^2 \tag{23}$$

The first two terms in Equation (20) are bound using uniform convergence for smooth non-negative losses, Theorem 1 in [SST10] and Bernstein's inequality as follows; with probability at least $1 - \beta/4$, we have,

$$\left|L(\widehat{h}; \mathcal{D}) - \widehat{L}(\widehat{h}; S_{\text{priv}})\right| + \left|L(\tilde{h}^*; \mathcal{D}) - \widehat{L}(\tilde{h}^*; S_{\text{priv}})\right|$$

$$= \tilde{O}\left(\sqrt{H}\mathfrak{R}_{n_{\text{priv}}}(\mathcal{H}) + \sqrt{\frac{B\log(8/\beta)}{n_{\text{priv}}}}\right)\left(\sqrt{\widehat{L}(\widehat{h}; S_{\text{priv}})} + \sqrt{L(\tilde{h}^*; \mathcal{D})}\right)$$

$$+ \tilde{O}\left(\mathfrak{R}_{n_{\text{priv}}}(\mathcal{H})^2 + \frac{B\log(8/\beta)}{n_{\text{priv}}}\right)$$

$$= \tilde{O}\left(\sqrt{H}\mathfrak{R}_{n_{\text{priv}}}(\mathcal{H}) + \sqrt{\frac{B\log(8/\beta)}{n_{\text{priv}}}}\right)\left(\sqrt{\widehat{L}(\tilde{h}^*; S_{\text{priv}})} + \sqrt{L(\tilde{h}^*; \mathcal{D})}\right)$$

$$+ \tilde{O}\left(H\mathfrak{R}_{n_{\text{priv}}}(\mathcal{H})^2 + \frac{B\log(8/\beta)}{n_{\text{priv}}}\right) + \tilde{O}\left(\frac{B(\text{fat}_{c\alpha}(\mathcal{H}) + \log(4/\beta))}{n_{\text{priv}}\epsilon}\right)$$

$$= \tilde{O}\left(\sqrt{H}\mathfrak{R}_{n_{\text{priv}}}(\mathcal{H}) + \sqrt{\frac{B\log(8/\beta)}{n_{\text{priv}}}}\right)\sqrt{L(\tilde{h}^*; \mathcal{D})}$$

$$+ \tilde{O}\left(H\mathfrak{R}_{n_{\text{priv}}}(\mathcal{H})^2 + \frac{B\log(8/\beta)}{n_{\text{priv}}}\right) + \tilde{O}\left(\frac{B(\text{fat}_{c\alpha}(\mathcal{H}) + \log(4/\beta))}{n_{\text{priv}}\epsilon}\right)$$

$$= \tilde{O}\left(\sqrt{H}\mathfrak{R}_{n_{\text{priv}}}(\mathcal{H}) + \sqrt{\frac{B\log(8/\beta)}{n_{\text{priv}}}}\right)\sqrt{L(h^*; \mathcal{D})}$$

$$+ \tilde{O}\left(H\mathfrak{R}_{n_{\text{priv}}}(\mathcal{H})^2 + \frac{B\log(8/\beta)}{n_{\text{priv}}}\right) + \tilde{O}\left(\frac{B(\text{fat}_{c\alpha}(\mathcal{H}) + \log(8/\beta))}{n_{\text{priv}}\epsilon}\right)$$

where the second equality follows from Equation (21), concavity of $x \mapsto \sqrt{x}$ and AM-GM inequality. The third equality follows concavity of $x \mapsto \sqrt{x}$ and Bernstein's inequality, the fourth follows from Equation (23) and AM-GM inequality.

Plugging the above, Equation (22) and Equation (21) into Equation (20) yields the following with probability at least $1 - \beta$,

$$L(\widehat{h}; \mathcal{D}) - L(h^*; \mathcal{D})$$

$$= \tilde{O}\left(\sqrt{H}\mathfrak{R}_{n_{\mathrm{priv}}}(\mathcal{H}) + \sqrt{H}\alpha + \sqrt{\frac{B\log(8/\beta)}{n_{\mathrm{priv}}}}\right)\sqrt{L(h^*; \mathcal{D})} \tag{24}$$

$$+ \tilde{O}\left(H\mathfrak{R}_{n_{\mathrm{priv}}}^2(\mathcal{H}) + H\alpha^2 + \frac{B\log(8/\beta)}{n_{\mathrm{priv}}} + \frac{B(\mathrm{fat}_{c\alpha}(\mathcal{H}) + \log(4/\beta))}{n_{\mathrm{priv}}\epsilon}\right) \tag{25}$$

This completes the first part of the theorem. For the second part, we proceed from Equation (25) onwards

$$L(\widehat{h}; \mathcal{D}) \leq L(h^*; \mathcal{D}) + \tilde{O}\left(\sqrt{H}\mathfrak{R}_{n_{\mathrm{priv}}}(\mathcal{H}) + \sqrt{H}\alpha + \sqrt{\frac{B\log(8/\beta)}{n_{\mathrm{priv}}}}\right)\sqrt{L(h^*; \mathcal{D})}$$

$$+ \tilde{O}\left(H\mathfrak{R}_{n_{\mathrm{priv}}}^2(\mathcal{H}) + H\alpha^2 + \frac{B\log(8/\beta)}{n_{\mathrm{priv}}} + \frac{B(\mathrm{fat}_{c\alpha}(\mathcal{H}) + \log(4/\beta))}{n_{\mathrm{priv}}\epsilon}\right)$$

$$\leq L(\widehat{h}^*; \mathcal{D}) + \tilde{O}\left(\sqrt{H}\mathfrak{R}_{n_{\mathrm{priv}}}(\mathcal{H}) + \sqrt{H}\alpha + \sqrt{\frac{B\log(8/\beta)}{n_{\mathrm{priv}}}}\right)\sqrt{L(\widehat{h}^*; \mathcal{D})}$$

$$+ \tilde{O}\left(H\mathfrak{R}_{n_{\mathrm{priv}}}^2(\mathcal{H}) + H\alpha^2 + \frac{B\log(8/\beta)}{n_{\mathrm{priv}}} + \frac{B(\mathrm{fat}_{c\alpha}(\mathcal{H}) + \log(4/\beta))}{n_{\mathrm{priv}}\epsilon}\right)$$

$$\leq \widehat{L}(\widehat{h}^*; S_{\mathrm{priv}}) + \tilde{O}\left(\sqrt{H}\mathfrak{R}_{n_{\mathrm{priv}}}(\mathcal{H}) + \sqrt{H}\alpha + \sqrt{\frac{B\log(8/\beta)}{n_{\mathrm{priv}}}}\right)\sqrt{\widehat{L}(\widehat{h}^*; S_{\mathrm{priv}})}$$

$$+ \tilde{O}\left(H\mathfrak{R}_{n_{\mathrm{priv}}}^2(\mathcal{H}) + H\alpha^2 + \frac{B\log(8/\beta)}{n_{\mathrm{priv}}} + \frac{B(\mathrm{fat}_{c\alpha}(\mathcal{H}) + \log(4/\beta))}{n_{\mathrm{priv}}\epsilon}\right)$$

where the second inequality follows form optimality of $h^*$, the third from uniform convergence, Theorem 1 in [SST10] and AM-GM inequality. This completes the proof.

$\square$

### D.3  Proof of Corollary 2

We use the result from [GRS18], restated as Theorem 10. Further, note that range bound on the hypothesis class is simply $R \leq \|\mathcal{X}\| \prod_{j=1}^{d} R_j$. Instantiating our general result Theorem 7 with the above together with the relation between fat-shattering dimension and Rademacher complexity (Theorem 9) yields the following excess risk bound,

$$L(\widehat{h}; \mathcal{D}) - \min_{h \in \mathcal{H}} L(h; \mathcal{D})$$

$$= O\left(\frac{G\|\mathcal{X}\|\left(\sqrt{2\log(2)M} + 1\right)\prod_{j=1}^{M} R_j}{\sqrt{n_{\mathrm{priv}}}} + \frac{B\sqrt{\log(4/\beta)}}{\sqrt{n_{\mathrm{priv}}}} + \frac{B\log(4/\beta)}{n_{\mathrm{priv}}\epsilon}\right)$$

$$+ \tilde{O}\left(\left(\frac{BG^2(\sqrt{2\log(2)M} + 1)^2\|\mathcal{X}\|^2\left(\prod_{j=1}^{M} R_j\right)^2}{n_{\mathrm{priv}}\epsilon}\right)^{1/3}\right).$$

where in the above, we set $\alpha = \left(\frac{B(\sqrt{2\log(2)M}+1)^2\|\mathcal{X}\|^2(\prod_{j=1}^{M} R_j)^2}{Gn_{\mathrm{priv}}\epsilon}\right)^{1/3}$.

The number of public samples then is

$$n_{\text{pub}} = O\left(\max\left(\frac{R^2 \log\left(2/\beta\right)}{\alpha^2}, \min\left\{m : \log^3(m)\mathfrak{R}_m^2(\mathcal{H}) \leq \alpha^2\right\}\right)\right)$$

$$= \tilde{O}\left(\|\mathcal{X}\|^2 \,(\prod_{j=1}^{M} R_j)^2 \max\left(\frac{\log\left(2/\beta\right)}{\alpha^2}, \frac{M}{\alpha^2}\right)\right)$$

$$= \tilde{O}\left((\|\mathcal{X}\| \,(\prod_{j=1}^{M} R_j))^{2/3}(n_{\text{priv}}\epsilon)^{2/3} M^{1/3} \log\left(2/\beta\right)\right).$$

### D.4 Proof of Corollary 3

Note that $R \leq D\,\|\mathcal{X}\|$. Further, we have [KST08, FGV17],

$$\mathfrak{R}_m(\mathcal{H}) = O\left(\frac{D\,\|\mathcal{X}\|}{m^{1/r}}\right).$$

Moreover, we have from Theorem 9, for any $\alpha > \mathfrak{R}_m(\mathcal{H})$,

$$\text{fat}_\alpha(\mathcal{H}) \leq \frac{4m\mathfrak{R}_m^2(\mathcal{H})}{\alpha^2}.$$

Choose $m = n_{\text{pub}}$ and $\alpha = (\log\left(n_{\text{pub}}\right))^{3/2}\mathfrak{R}_{n_{\text{pub}}}(\mathcal{H})$, to get that $\text{fat}_\alpha(\mathcal{H}) = \tilde{O}\left(n_{\text{pub}}\right)$. Plugging this in Theorem 7, we get,

$$L(\widehat{w};\mathcal{D}) - \min_{w\in\mathcal{W}} L(w;\mathcal{D}) = O\left(\frac{GD\,\|\mathcal{X}\|}{n_{\text{priv}}^{1/r}} + \frac{GD\,\|\mathcal{X}\|\sqrt{\log\left(4/\beta\right)}}{\sqrt{n_{\text{priv}}}} + \frac{B\sqrt{\log\left(4/\beta\right)}}{\sqrt{n_{\text{priv}}}}\right)$$

$$+ \tilde{O}\left(GD\,\|\mathcal{X}\|\left(\frac{n_{\text{pub}}}{n_{\text{priv}}\epsilon} + \frac{1}{n_{\text{pub}}^{1/r}} + \frac{\log\left(4/\beta\right)}{n_{\text{priv}}\epsilon}\right)\right)$$

$$= \tilde{O}\left(GD\,\|\mathcal{X}\|\left(\frac{1}{n_{\text{priv}}^{1/r}} + \frac{\sqrt{\log\left(4/\beta\right)}}{\sqrt{n_{\text{priv}}}} + \frac{\log\left(2/\beta\right)}{(n_{\text{priv}}\epsilon)^{\frac{1}{r+1}}} + \frac{\log\left(4/\beta\right)}{n_{\text{priv}}\epsilon}\right)\right)$$

$$+ O\left(\frac{B\sqrt{\log\left(4/\beta\right)}}{\sqrt{n_{\text{priv}}}}\right),$$

where in the last step, we plug in $n_{\text{pub}} = O\left((n_{\text{priv}}\epsilon)^{r/(1+r)}\log\left(2/\beta\right)\right)$, yielding $\alpha = O\left(\frac{D\|\mathcal{X}\|}{(n_{\text{priv}}\epsilon)^{\frac{1}{r+1}}}\right)$. The number of public samples simplifies as,

$$n_{\text{pub}} = O\left(\max\left(\frac{R^2 \log\left(2/\beta\right)}{\alpha^2}, \min\left\{m : \log^3(m)\mathfrak{R}_m^2(\mathcal{H}) \leq \alpha^2\right\}\right)\right)$$

$$= \tilde{O}\left((n_{\text{priv}}\epsilon)^{\frac{2}{r+1}}\log\left(2/\beta\right)\right).$$

which is satisfied from our choice since $r \geq 2$.

### D.5 Additional Results

**Corollary 4.** *In the setting of Theorem 7 together with* $\mathcal{X} = \left\{x \in \mathbb{R}^d : \|x\| \leq \|\mathcal{X}\|\right\}$ *and* $\mathcal{H} = \left\{x \mapsto \langle w, x\rangle, x \in \mathcal{X}, \|w\| \leq D\right\}$. *With* $\alpha = \left(\frac{D^2\|\mathcal{X}\|^2}{n_{priv}\epsilon}\right)^{1/3}$ *and* $n_{pub} = \tilde{O}\left((D\,\|\mathcal{X}\|)^{2/3}(n_{priv}\epsilon)^{2/3}\log\left(2/\beta\right)\right)$, *with probability at least* $1 - \beta$,

$$L(\widehat{h};\mathcal{D}) - \min_{h\in\mathcal{H}} L(h;\mathcal{D}) = \tilde{O}\left(\frac{GD\,\|\mathcal{X}\|}{\sqrt{n_{priv}}} + G\left(\frac{D^2\,\|\mathcal{X}\|^2}{n_{priv}\epsilon}\right)^{1/3} + \frac{B\sqrt{\log\left(4/\beta\right)}}{\sqrt{n_{priv}}} + \frac{B\log\left(4/\beta\right)}{n_{priv}\epsilon}\right).$$

*Proof.* In this setting, we have that $R \leq D \|\mathcal{X}\|$. Further, it is known [KST08],

$$\text{fat}_\alpha(\mathcal{H}) = O\left(\frac{D^2 \|\mathcal{X}\|^2}{\alpha^2}\right), \qquad \mathfrak{R}_n(\mathcal{H}) = O\left(\frac{D \|\mathcal{X}\|}{\sqrt{m}}\right).$$

Plugging this in Theorem 7, we get,

$$L(\widehat{h}; \mathcal{D}) - \min_{h \in \mathcal{H}} L(h; \mathcal{D})$$

$$= O\left(\frac{GD \|\mathcal{X}\|}{\sqrt{n_{\text{priv}}}}\right) + O\left(\frac{B\sqrt{\log(4/\beta)}}{\sqrt{n_{\text{priv}}}}\right) + \tilde{O}\left(\frac{\min(B, GD \|\mathcal{X}\|)D \|\mathcal{X}\|}{\alpha^2 n_{\text{priv}}\epsilon}\right)$$

$$+ 2G\alpha + O\left(\frac{B\log(4/\beta)}{n_{\text{priv}}\epsilon}\right)$$

$$= \tilde{O}\left(\frac{GD \|\mathcal{X}\|}{\sqrt{n_{\text{priv}}}} + G\left(\frac{D^2 \|\mathcal{X}\|^2}{n_{\text{priv}}\epsilon}\right)^{1/3} + \frac{B\sqrt{\log(4/\beta)}}{\sqrt{n_{\text{priv}}}} + \frac{B\log(4/\beta)}{n_{\text{priv}}\epsilon}\right).$$

where in the last step, we plug in $\alpha = \left(\frac{D^2\|\mathcal{X}\|^2}{n_{\text{priv}}\epsilon}\right)^{1/3}$. The number of public samples simplifies as,

$$n_{\text{pub}} = O\left(\max\left(\frac{R^2 \log(2/\beta)}{\alpha^2}, \min\left\{m : \log^3(m)\mathfrak{R}_m^2(\mathcal{H}) \leq \alpha^2\right\}\right)\right)$$

$$= \tilde{O}\left(D^2 \|\mathcal{X}\|^2 \max\left(\frac{\log(2/\beta)}{\alpha^2}, \frac{1}{\alpha^2}\right)\right)$$

$$= \tilde{O}\left((D \|\mathcal{X}\|)^{2/3}(n_{\text{priv}}\epsilon)^{2/3} \log(2/\beta)\right).$$

$\square$

