# OpenReview forum: "Public-data Assisted Private Stochastic Optimization: Power and Limitations"
_NeurIPS.cc/2024/Conference — NeurIPS 2024 poster_

### Official Review · Reviewer_9k4B · 2024-07-12

**Soundness:** 3
**Presentation:** 1
**Contribution:** 2
**Rating:** 5
**Confidence:** 3

**Summary:**

The paper presents some new lower bounds for public-data assisted private SCO, when some public examples are available, either with or without labels.
In the unlabeled case, a simple algorithm for GLM assisted with unlabeled public examples is presented and analyzed (and it is shown that this removes a dimension dependent term, similar to previous results).

**Strengths:**

- The problem is important.
- The new results shed additional light on the limits of public-data assisted private SCO.
- In particular, they partially closes the gap between lower and upper bounds, in certain regimes of $d$ and $n_{pub}$.
- There is good commentary following most of the results.

**Weaknesses:**

- The presentation is poor. Little effort was made to summarize the main results (and how they improve on previous bounds) in an easily accessible form. This could be remedied with a table summarizing assumptions, new and existing results, conditions on $n_{priv}, n_{pub}, \epsilon, d$, ranges in which the bounds are tight, etc.
- For example, it was unclear from the abstract and introduction in which regimes the new bounds are tight. This is only discussed later in the text. Similarly, there are boundedness assumption that should have been clarified early on.
- In Theorems 3, 6, 7, it seems there are missing conditions on $n_{pub}$. For example, according to the theorem statement, nothing prevents one from taking $n_{pub} = 1$, or even 0, and it seems one would still get the improved bound. This should not be the case. Please clarify in the rebuttal.
- The significance of Theorem 3 is unclear. Similar algorithms and similar improvements of dimension dependence were obtained in other works. The authors mention line 245 that their result avoids "many of the strong assumptions seen in previous work" without giving any details. What are these assumptions and how does Theorem 3 compare to existing results?
- Corollaries 2 and 3 are frankly hard to interpret, and their significance is unclear to me. Maybe more effort is needed in this section.
- The general message of the paper, that "more sophisticated attempts at leveraging public data will yield no benefit" (in the labeled case), should be toned down a bit. While the new bounds indeed show some fundamental limits, these come under assumptions, and improvements were in fact shown for certain classes of problems like linear regression, as in some of the papers that were cited lines 28-29, (most of these were cited without discussion unfortunately).

Minor:
- What do the authors mean by "the non trivial regime $n_{pub} = \Theta(n)$ and $n_{pub} = o(n)$"? Do you mean either of these is true? (if so, it would be clearer to use or, not and).
- There are minor formatting issues like a broken reference line 359 and some typos.
- In the discussion following Cor.3, how do the $log d$ terms appear?

**Questions:**

Please see questions in the weaknesses.

**Limitations:**

As mentioned above, the paper should be clearer about its assumptions, regimes in which its bounds are tight, etc.

---

> ### Author Rebuttal · Authors · 2024-08-03
>
> * *Presentation:*
>
> We will add a table to our updated version as you suggest and clarify tightness of the rates. Please see the global response PDF for this table.
>
> * *Missing conditions on $n_\text{pub}$ :*
>
> Theorems 3, 6, and 7 claim the rate is achievable given access to *some* number of public samples which is at most $n_\text{pub} = \tilde{O}(n_{\text{priv}}\epsilon)$. We will clarify the phrasing of these theorems to make this more clear.
>
> * *The significance of Theorem 3 is unclear. Similar algorithms and similar improvements of dimension dependence were obtained in other works. The authors mention line 245 that their result avoids "many of the strong assumptions seen in previous work" without giving any details. What are these assumptions and how does Theorem 3 compare to existing results?*
>
> With regards to the [[PHYS24]](https://arxiv.org/pdf/2306.03962) paper we cite, in addition to the assumptions we make, [PHYS24] additionally requires that the underlying distribution satisfies both a large margin condition and a low rank assumption (see Definition 3 and Theorem 1 of their paper). Further, they also only provide a statement for the privacy regime $\epsilon = o(1)$. We will add these details to the revision.
>
> * *Corollaries 2 and 3 are frankly hard to interpret, and their significance is unclear to me. Maybe more effort is needed in this section.*
>
> Corollary 2 shows that with enough public data, it is possible to privately learn feed-forward neural networks without paying any explicit penalty for the network width (the rate only scales with the norm of the weights).
> Corollary 3 shows that (with access to public data) it is possible to obtain dimension-independent rates for constrained non-Euclidean GLMs. In both cases, such a width/dimension independence was previously unknown. We will add such comments to the revision.
>
> * *The general message of the paper, that "more sophisticated attempts at leveraging public data will yield no benefit" (in the labeled case), should be toned down a bit. While the new bounds indeed show some fundamental limits, these come under assumptions, and improvements were in fact shown for certain classes of problems like linear regression, as in some of the papers that were cited lines 28-29, (most of these were cited without discussion unfortunately).*
>
> As with all theoretical works, our results come with certain assumptions, and our statements are made with respect to those assumptions. The class of problems we study for our lower bounds, Lipschitz losses over compact parameter space, are one of the most commonly studied classes in the DP optimization literature. Further, we do not believe the overall tone of our paper suggests that more sophisticated algorithms are useless in other settings. To the contrary, half our paper is devoted towards showing that in a slightly different setting (unlabeled public data) more sophisticated algorithms *are* useful.
>
> * *What do the authors mean by "the non trivial regime  $n_{\text{priv}} = \Theta(n)$ and $n_{\text{pub}} = o(n)$"? Do you mean either of these is true? (if so, it would be clearer to use or, not and).*
>
> We mean "and" for this statement. The regime $n_{\text{priv}} = \Theta(n)$ and $n_{\text{pub}} = o(n)$ is the only non-trivial regime, at least if one is interested in asymptotics. This is because if either condition is not true, one can discard the private dataset and still have a dataset of size $\Theta(n)$; i.e., one does not have to worry about privacy. This is obvious if $n_{\text{pub}}=o(n)$ is not true. If $n_{\text{priv}}=\Theta(n)$ is not true, then it must also be the case that $n_{\text{pub}} = \Theta(n)$ since by definition $n = n_{\text{pub}} + n_{\text{priv}}$. We will elaborate in the revision.
>
> * *In the discussion following Cor.3, how do the $\log(d)$ terms appear?*
>
>  The $\log{d}$ term appears from the choice of the function $\Delta(w) = \frac{\log{d}}{2}\Vert w \Vert_{1+1/\log{d}}$ -- see discussion in line 366 in the paper. This choice is standard in the $(\ell_1,\ell_\infty)$-case -- see referenced work [FGV17]. We will revise the Corollary and discussion following it to make it clear.

---

> > ### Comment · Reviewer_9k4B · 2024-08-10
> >
> > Thank you for the response and clarifications.
> >
> > - The discussion about Corollary 2 and 3 is helpful and should be expanded in the paper.
> >
> > > Theorems 3, 6, and 7 claim the rate is achievable given access to some number of public samples
> >
> > - Do you mean that there should be a lower bound on $n_{pub}$ (in addition to the upper bound) that is missing from the statements of these theorems? Please clarify.
> >
> > - Regarding the non trivial regime: It appears there is a typo in the paper that led to my misunderstanding (both conditions applied to $n_{pub}$, instead of $n_{pub},n_{priv}$...) It is clear now (but please make sure to correct the typo).
> >
> > - Regarding the message of the paper: are the authors willing to rephrase some of the statements made in the paper? For example, the abstract claims that "the simple strategy of either treating all data as private or discarding the private data, is optimal". This should be properly qualified. Similarly for the statement "more sophisticated attempts at leveraging public data will yield no benefit".

---

> > > ### Author Response · Authors · 2024-08-13
> > >
> > > Thank you for your response. We will add the discussion about Corollary 2 and 3 to the paper. We will also fix the typo regarding $n_{pub}$.
> > >
> > > To further clarify Theorems 3, 6, and 7, there are alternative ways to the state the condition on $n_{pub}$. For example,  we can either state it as 1) "There exists some​ number $n_{pub} = \tilde{O}(n_{priv} \epsilon)$ such that ..."  or  2) "There is a constant c > 0 s.t. for any​ $n_{pub} \geq c n_{priv} \epsilon$, we have ..."  To be clear, our intention for what is written in the paper is *not* an upper bound of the form "for any $n_{pub} = \tilde{O}(n_{priv} \epsilon)$ but rather an upper bound on the number of samples the algorithm would ever need. To make it more clear, we can replace what is in the paper with "Then there exists some  $n_{pub} = \tilde{O}(n_{priv} \epsilon)$..." or "Then there exists a universal constant c such that for any $n_{pub} \geq c n_{priv} \epsilon$..."
> > >
> > > With regards to the message of the paper, we are happy to be more explicit about our claims in the abstract. Specifically, we can say that our claim applies to the problem of stochastic convex optimization of Lipschitz functions and our claim is made with respect to asymptotic rates. Please let us know if there are other assumptions worth enumerating in the abstract.

---

### Official Review · Reviewer_VXJL · 2024-07-12

**Soundness:** 4
**Presentation:** 3
**Contribution:** 3
**Rating:** 7
**Confidence:** 3

**Summary:**

The paper investigates a public-data-assisted differential privacy problem. Firstly, for labeled public data, the author introduces a novel mean estimation lower bound of $\tilde{\Omega}\left(\min \left{\frac{1}{\sqrt{n_{\mathrm{pub}}}}, \frac{1}{\sqrt{n}}+\frac{\sqrt{d}}{n \epsilon}\right}\right)$. Secondly, when considering unlabeled public data with $n_{\text {pub }} \geq n_{\text {priv }}$, the paper presents a dimension-independent rate of $\tilde{O}\left(\frac{1}{\sqrt{n_{\text {priv }}}}+\frac{1}{\sqrt{n_{\text {priv }} \epsilon}}\right)$ given $\tilde{O}\left(n_{\text {priv }} \epsilon\right)$ unlabeled public data. Additionally, the results for unlabeled public data are extended to general hypothesis classes with bounded fat-shattering dimensions.

**Strengths:**

1. The paper provides a technically solid and well-structured analysis of the public-data-assisted differential privacy problem. The most intriguing part is the Private Supervised Learning with Unlabeled Public Data. In this setting, the paper leverages $n_{\text {pub }} \geq n_{\text {priv }}$, showing a dimension-independent rate and demonstrating that an increasing number of unlabeled public data cannot improve the results because unlabeled public data can only reveal information about the marginal distribution.

2. The emergence of the dimension-independent rate from "public data can be used to identify a low-dimensional subspace, which under the appropriate metric acts as a cover for the higher-dimensional space," is particularly interesting. Could the author provide more explanations and insights on this?

3. Algorithm 1 projects the private data to an orthogonal space, derives the solution, and then projects back to the original space to obtain the result. This approach is novel to me, but I have some questions below.

**Weaknesses:**

N/A

**Questions:**

For Algorithm 1, why does the author perform the projection before applying the differential privacy subroutine? What is the function of the projection here, or how does it help the result? I am not sure if it is because the author uses the unlabeled dataset to construct a data space.

**Limitations:**

Refers to the weakness and questions.

---

> ### Author Rebuttal · Authors · 2024-08-03
>
> * *Response to Questions:*
>
> Differentially private subroutines often incur some penalty proportional to the problem dimension, $d$. Thus, at a high level, the reason for performing the projection before applying the DP subroutine is to ensure that the penalty does not scale with $d$.
> Indeed, we show that one can reduce the dimension of the problem (using public data) to the extent that, no dependence on $d$ (the original dimension) shows up in the final rate. This is because the effective dimension after projection does not exceed $n_{pub}$.

---

> > ### Comment · Reviewer_VXJL · 2024-08-09
> >
> > Thank you for the clarification. I will maintain my current score.

---

### Official Review · Reviewer_42Zi · 2024-07-15

**Soundness:** 3
**Presentation:** 3
**Contribution:** 3
**Rating:** 6
**Confidence:** 2

**Summary:**

This paper studies the effectiveness of using public data to assist private convex optimization. Private convex optimization gives the solver query access to a convex function $f:W\times X \to \mathbb{R}$ and samples $S$ drawn i.i.d. from some unknown distribution $D$. The solver is asked to find some point $w\in W$ that minimizes the population risk $\mathbb{E}_{x\sim D}[f(w,x)]$, but to do so in a manner that is private with respect to the sample $S$. This is an important problem that has received much interest in the community. The notion of using public data that one expects has been drawn from the same distribution has been explored earlier, and the authors continue this line of work.

The main contributions of this work are:
1. Stronger lower bounds than prior work showing that in the general setting even for approximate DP there is a dependence on the dimension $d$ of the sample space $X = \mathbb{R}^d$ (typically one assumes data is drawn from some bounded $d$-dimensional ball in the DP setting). They also complement prior work and extent a lower bound for the pure DP setting that held previously only for a limited range of values. These lower bounds show that the naive approach of either treating all data (public and private) as private or ignoring the private data and using SCO on public data is asymptotically optimal.
2. A new algorithm for the unlabelled setting showing that one can gainfully use unlabelled public data to perform dimension reduction for labelled private data and achieve a risk bound that has no dependence on the ambient dimension. This is in contrast to the labelled setting where their lower bounds showed a dependence on the ambient dimension $d$.

**Strengths:**

1. I think this problem is well-motivated and of interest to the DP community.
2. The asymptotic separation between the labelled and unlabelled settings is interesting, and the improvement of lower bounds compared to prior work is also good to see.
3. The paper is in general well-written and easy to follow.

**Weaknesses:**

1. I think that in this case it becomes important to understand the exact constants that occur in the upper bound in the labelled case - even if asymptotically one does not hope to see any improvement over the naive strategies, I would imagine that in practice one still does gain some improvement in the public case. Maybe this point could be better made.
2. Although the analyses seem to be formal and strong (and relatively assumption free), there are not necessarily any new high-level ideas in this work (if you would like to emphasize any new highlights or unexpected outcomes that would be great to see though). The technical work is still interesting however.

**Questions:**

There is one point that I don't understand, I think I might be confused about the nature of the results. In principle, if one treats public labelled data as unlabelled and attempts to follow a similar approach of using them to learn a good lower dimensional representation of the labelled private data, is it easy to see why this does not lead to an asymptotic improvement over ignoring the public data entirely, or treating it as private?

**Limitations:**

No additional limitations need to be discussed.

---

> ### Author Rebuttal · Authors · 2024-08-03
>
> * *Response to Weakness 1:*
>
> We completely agree exact constants are important here, and we will make a note to better emphasize this. However, we do not believe this devalues the importance of characterizing the asymptotic rates,   as we do in our paper. Most of the existing understanding of (private and non-private) optimization is in terms of asymptotic rates.
>
> * *Response to Weakness 2:*
>
> Many of the novel ideas in our paper are more technical in nature. For example, in our lower bound proof, while we leverage the pre-existing framework of fingerprinting codes, our proof involves several novel steps to overcome the challenges of applying this framework to PA-DP settings. For example, we observe it is crucial to use fingerprinting distributions with small mean and also rely on a clipping analysis when getting the $\sqrt{\log(1/\delta)}$ improvement in the lower bound. This analysis is different from the way fingerprinting codes have been used previously. We include further discussion of these novelties in Appendix B.2, but did not have space to include this in the main body. We can include this in the final version using the extra space given.
>
> With regards to the proof ideas in the unlabelled public data section, while it indeed bears similarities to existing work of [[ABM19]](https://dl.acm.org/doi/pdf/10.5555/3454287.3455215), which we have acknowledged in the paper, there are several novel components in the analysis; for instance, the analysis of the size of the cover required. We perform this analysis by posing it as another supervised learning problem in the realizable setting with squared loss.
> Using optimistic rates for the squared loss [[SST08]](https://papers.nips.cc/paper_files/paper/2010/file/76cf99d3614e23eabab16fb27e944bf9-Paper.pdf),
> we achieve fast $1/n$ rate for this intermediate learning problem, which is crucial in achieving optimal rates. This is in contrast to the analysis in [[ABM19]](https://dl.acm.org/doi/pdf/10.5555/3454287.3455215),
> which is done via VC-dimension based counting arguments.
>
> * *Response to Questions:*
>
> This could lead to an improvement over ignoring the *public* data entirely, but will not lead to an improvement over ignoring the *private* data entirely. That is, one incurs a $\frac{1}{\sqrt{n_{\text{pub}}}}$ penalty due to the error in the subspace approximation made using $n_{\text{pub}}$ points. Note that this is never better than the $O(\frac{1}{\sqrt{n_{\text{pub}}}})$ rate achieved by ignoring the private dataset and running some optimal non-private optimization algorithm.

---

> > ### Comment · Reviewer_42Zi · 2024-08-14
> > **Response to rebuttal**
> >
> > Thank you for your rebuttal! I will keep my score.

---

### Author Rebuttal · Authors · 2024-08-03

See the attached pdf for the lower bound table that will be added the revision.

---

### Decision · Program_Chairs · 2024-09-25

**Decision:**

Accept (poster)

**Comment:**

The paper considers the problem of differentially private stochastic convex optimization when there is access to public data for which privacy needn't be ensured. The paper provides improved and nearly optimal lower bounds in various useful settings. In the case of unlabelled public data the paper also shows a dimension improvement.

Overall the results of the paper were appreciated by the reviewers with only one major point of contention which is the writing. Many points have been raised by the reviewers including better presentation of the results comparison (which the authors have proposed to add) and better presentation of the required conditions on n_pub, i.e. using an O() notation or explicitly stating npub > c.n for some c. My personal recommendation is to use the latter as I agree with the reviewer that the former can lead to a different conclusion to a less careful reader. There are also other discussions and clarifications reviewers have suggested to add.

Since fixable presentation issues are the only main issue with the paper, I recommend marginal acceptance under the hope that the authors address these before the final version.